# Relation Models of Surface Parameters and Backscattering (or Radiation) Fields as a Tool for Solving Remote Sensing Problems

**Kseniia Nezhalska [1], Valerii Volosyuk [1], Kostiantyn Bilousov [2], Denys Kolesnikov [1],\***  **and Glib Cherepnin [1]**

1   Aerospace Radio-Electronic Systems Department, National Aerospace University "Kharkiv Aviation Institute", 61070 Kharkiv, Ukraine; k.nezhalska@khai.edu (K.N.)
2   Spacecraft, Measuring Systems and Telecommunications Department, Yuzhnoye SDO, 49000 Dnipro, Ukraine
\*   Correspondence: d.kolesnikov@khai.edu

**Abstract:** In this paper, an analysis of existing models for describing surfaces of various types is performed, and the possibilities of their application at the level of mathematical modeling are analyzed. Moreover, due to the large number of models and the complexity of selecting the appropriate model, e.g., when conducting a practical experiment, an algorithm for choosing a specific model depending on the initial data is proposed. According to the algorithm, a software prototype that implements this algorithm (written in Python) is proposed.

**Keywords:** brightness temperature; remote sensing; surface model; permittivity; conductivity

## 1. Introduction

In the modern world, the tasks of researching, studying, and saving our planet, in view of climate change and other environmental and anthropogenic problems, are very important. One of the most effective modern methods of earth surface exploration is through the use of remote sensing systems. In modern science and technology, remote sensing and measuring is one of the main methods of analysis in many fields: natural science [1–8], agriculture [9–15], climatology [16–21], etc. At the same time, models of these surfaces are necessary to relate the parameters of the studied surfaces (or objects) to the electromagnetic fields that these surfaces radiate or reflect. When preparing experimental studies, or interpreting the results of such experimental studies, theoretical modeling is important, and when determining the potential capabilities of various remote measurements, knowledge of the proper model of the field and parameter relations is one of the conditions that make the results of a study efficient and precise. Modern publications currently do not provide a comprehensive approach to creating, analyzing, or using models to describe actual ground surfaces. According to the received signals, radio engineering systems can be divided into either active or passive systems. The signals received by the system are related to the surface parameters via the radar cross-section or brightness temperature. There are many stand-alone articles and studies, but there is no generally accepted classification or structural approach to the model selection task. However, choosing the right mathematical models for these characteristics is the key to efficiency both in practical experiments and in modeling or interpreting the results. To simplify the optimal model selection process, this paper analyzes the existing mathematical relations and proposes both their classification and also an algorithm for model selection based on known initial data. A prototype of a software application that implements this algorithm has also been created.

## 2. Materials and Methods

One of the areas of remote sensing research that has been developing rapidly in recent years is the development and application of remote sensing systems. All radio frequency systems can be divided into two major classes: active and passive. Passive systems receive the surface's own thermal radiation, while active systems irradiate the surface or object

and receive a reflected signal, which contains information about the object parameters. Due to the different nature of electromagnetic waves, the characteristics that relate these waves to surface parameters are also different. In a passive case, such a characteristic is the brightness temperature $T_{Br}$, while in an active case, it is a radar cross-section (RCS) $\sigma^2$.

All models for surface description can be divided into being either electrodynamic (mathematical) [22,23] or empirical (practical) [24–40]. Mathematical models are well developed based on the general laws of electrodynamics and have been applied to remote sensing problems for a long time. Practical models reflect the results of specific experiments and are constantly being improved and developed.

For both electrodynamic and empirical models, specific conditions and limitations exist for their application, in particular in terms of operating frequencies f (wavelengths $\lambda$), covering types (with or without vegetation), and geometric characteristics: the root mean square heights of roughness $\sigma_h$ (or spatial heights of roughness $h(x, y)$), radius of roughness curvature $R_K$, and $\updownarrow$, which is the radius of roughness correlation.

Nowadays, in order to conduct a practical experiment or to simulate a remote sensing experiment and take into account the parameters of the studied surface, it is necessary to find a valid model. This, due to the large number of options, the conditions of their application, and the connection, in particular with the type of system used, is quite a complex, time-consuming, and laborious task.

This paper proposes to classify the existing surface models as follows:

1.  By classifying the type of systems that can be used (active, passive).
2.  By classifying the obtaining method (electrodynamic, empirical).
3.  By classifying the type of described surface (ground, water).
4.  By classifying the type of covering (ground surface without vegetation, surface with vegetation below a defined level, surface with high vegetation, smooth water surface, water surface with foam, surface covered with snow or ice).
5.  By classifying the ability to estimate atmospheric parameters (taking into account the atmosphere impact, along with the ability to estimate atmospheric parameters).

Table 1 shows the classification of surface model types, with descriptions, that are available in the public domain.

**Table 1.** Surface model classification.

| Type | Frequency Range (Wavelengths) | Conditions of Use |
|---|---|---|
| | Brightness temperature | |
| | Electrodynamic | |
| Flat | – | $h(x, y) = 0$, $\sigma_h \approx 0$ |
| Flat with the atmosphere | – | $h(x, y) = 0$, $\sigma_h \approx 0$ |
| Small-scale | – | $\|h(x, y)\| \ll \lambda$, $\dfrac{\partial h(x, y)}{\partial x} \ll 1$, $\dfrac{\partial h(x, y)}{\partial y} \ll 1$, $\sigma_h \leq \dfrac{\lambda}{20}$ |
| Large-scale | – | $R_{Kx} \gg \lambda, R_{Ky} \gg \lambda$, $\sigma_h \geq \lambda$ |
| Two-scale | – | $\sigma_{h1} \geq \lambda$, $\sigma_{h2} \ll \lambda$ |
| | Empirical | |
| Sea surface with foam | 9.3–34 GHz (8.8 mm—3.2 cm) | For a sea surface with foam (excluding atmospheric illumination), including wind speed |
| $\tau - \omega$ model | 4–8.8 GHz (3.4–7.5 cm) | Vegetation is an equally absorbing and backscattering layer over the soil surface |
| $Q_p$ model | 6.9–36.5 GHz (8.3 mm–4.3 cm) | Surface without vegetation |

**Table 1.** *Cont.*

| Type | Frequency Range (Wavelengths) | Conditions of Use |
|---|---|---|
| Regression model | 22.2–37.5 GHz (0.8–1.35 cm) | To estimate the moisture content of a cloudless atmosphere |
| "Meteor" regression model | 37.5 GHz (0.8 cm) | To estimate the moisture content of the atmosphere and clouds at a sight angle $\theta = 30°$ |
| "Nimbus 5" model | $\lambda_1 = 0.96$ cm, $\lambda_2 = 1.35$ cm—operating wavelengths, 31.25 GHz, 22.22 GHz | To estimate the moisture content of the atmosphere and clouds when sighting in nadir, $\theta = 0°$ |
| "Seasat" model | $\lambda_1 = 0.81$ cm, $\lambda_2 = 1.43$ cm, $\lambda_3 = 1.67$ cm $\lambda_4 = 2.8$ cm, $\lambda_5 = 4.55$ cm—operating wavelengths, frequencies 37.05 GHz, 20.98 GHz, 17.96 GHz, 10.71 GHz, 6.593 GHz | To measure the near-surface wind speed $v_{ns} < 7$ m/s, thermodynamic temperature $T_0$ [K], atmosphere's moisture content Q [mg/cm$^2$], and cloud moisture content W [mg/cm$^2$] |
| Radar Cross-Section (RCS) | | |
| Electrodynamic | | |
| Flat | – | $h(x,y) = 0$, $\sigma_h \approx 0$ |
| Small-scale | – | $\|h(x,y)\| \ll \lambda$, $\dfrac{\partial h(x,y)}{\partial x} \ll 1, \dfrac{\partial h(x,y)}{\partial y} \ll 1$, $\sigma_h \leq \dfrac{\lambda}{20}$ |
| Large-scale | – | $R_{Kx} \gg \lambda, R_{Ky} \gg \lambda$, $\sigma_h \geq \lambda$ |
| Two-scale | – | $\sigma_{h1} \geq \lambda$, $\sigma_{h2} \ll \lambda$ |
| Empirical | | |
| Exponential model | Frequency 3...100 GHz Wavelength 0.003–0.1 m | Quasi-smooth surfaces, rough surfaces with and without vegetation, as well as snow and anthropogenic areas Grazing angle $\psi \leq 30°$, $\psi = \dfrac{\pi}{2} - \theta$, $\theta$—angle of incidence |
| Oh's model | – | $0.1 < k \cdot \sigma_h < 6.0$, $2.6 < k \cdot \updownarrow < 19.7$, moisture content $0.09 < m < 0.31$ |
| Surface with vegetation | Frequency 1–18 GHz Wavelength 0.017–0.3 m | Surface with vegetation |
| Dubois model | Frequency 1.5–11 GHz Wavelength 0.027–0.2 m | Surfaces without vegetation, sighting angles from 30 to 65 degrees The normalized radar cross−sections ratio is $\sigma^0_{VV} \geq \sigma^0_{HH}$ $\sigma_h$ from 0.3 cm to 3 cm Angles of incidence are $\theta = 30°\ldots65°$ |
| Model with cylindrical reflectors | – | A surface that can be represented as a set of cylindrical reflectors |
| Integral equation model | – | Soils without vegetation at high root mean squares of roughness values. $k\sigma_h \ll 1, k = \dfrac{2\pi}{\lambda}$ |
| Model with a near-surface wind | – | $\varphi = 30, 40, 50$ deg is the angle with respect to the direction opposite to the wind vector |
| Shi's model-algorithm | – | Surface without vegetation |
| Surface model with snow | – | Surface covered with snow |

### 3. Results

#### 3.1. Electrodynamic Models of Brightness Temperature

It is typical (Figure 1) for a passive location that the observed object itself emits or re-emits natural radio-thermal signals from other hot objects (e.g., the Sun).

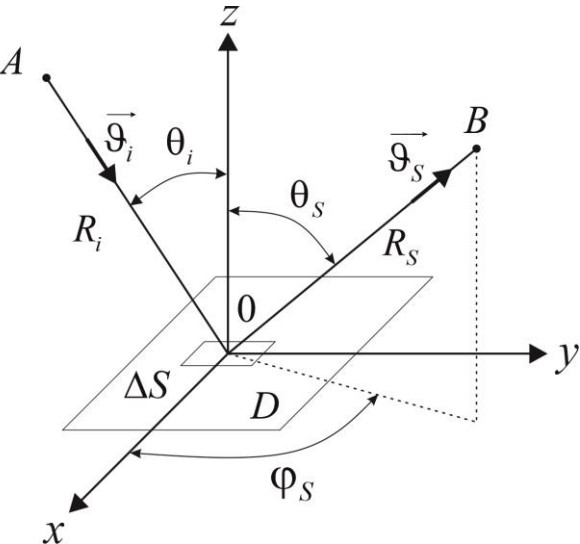

**Figure 1.** Surface radiolocation geometry.

By dividing the entire radiation-receiving area into main and side lobes and calculating the mean antenna temperatures in these areas, we obtain one of the fundamental formulas of radiometry:

$$T_A = T_\Gamma \eta (1 - \beta) + T_\sigma \eta \beta + T_0 (1 - \eta), \tag{1}$$

where $T_\Gamma$ is the radiating object's apparent temperature, smoothed (averaged) by the main lobe of the radiation pattern,

$$\beta = \frac{\int\limits_{\Omega_{side}} G_{rec}(\vec{\vartheta})}{\int\limits_{\Omega=4\pi} G_{rec}(\vec{\vartheta}) d\Omega} \tag{2}$$

is the scattering factor of the receiving antenna; $T_\sigma$ is the brightness temperature of the averaged background radiation from the side and back lobes; $\eta$ is the aperture efficiency; and $T_0$ is the thermodynamic temperature of the antenna–waveguide tract.

The apparent temperature of the radiating object includes the true radio brightness temperature and the temperature caused by the radiation illumination of the atmosphere, clouds, the Sun, etc.

#### 3.1.1. Electrodynamic Model of Flat Surface

The main characteristics of the flat surface model (Figure 2) are the Fresnel reflection coefficients for oscillations of plane waves with horizontal polarizations [22,23],

$$\dot{K}_{f\,H} = \frac{\dot{E}_{0\,ref}}{\dot{E}_{0\,inc}} = \frac{\sqrt{\dot{\varepsilon}_1} \cos\theta_1 - \sqrt{\dot{\varepsilon}_2 - \dot{\varepsilon}_1 \sin^2\theta_1}}{\sqrt{\dot{\varepsilon}_2} \cos\theta_1 + \sqrt{\dot{\varepsilon}_2 - \dot{\varepsilon}_1 \sin^2\theta_1}} \tag{3}$$

and vertical polarization,

$$\dot{K}_{f\,V} = \frac{\dot{\varepsilon}_2 \cos\theta_1 - \sqrt{\dot{\varepsilon}_1}\sqrt{\dot{\varepsilon}_2 - \dot{\varepsilon}_1 \sin^2\theta_1}}{\dot{\varepsilon}_2 \cos\theta_1 + \sqrt{\dot{\varepsilon}_1}\sqrt{\dot{\varepsilon}_2 - \dot{\varepsilon}_1 \sin^2\theta_1}}, \tag{4}$$

where $\dot{E}_{0\ \text{ref}}$, $\dot{E}_{0\ \text{inc}}$ are the reflected and incident field; $\dot{\varepsilon}_1, \dot{\varepsilon}_2$ are the dielectric constants of medium 1 and medium 2 (Figure 2); and $\theta_1$ is the sensing angle (Figure 2).

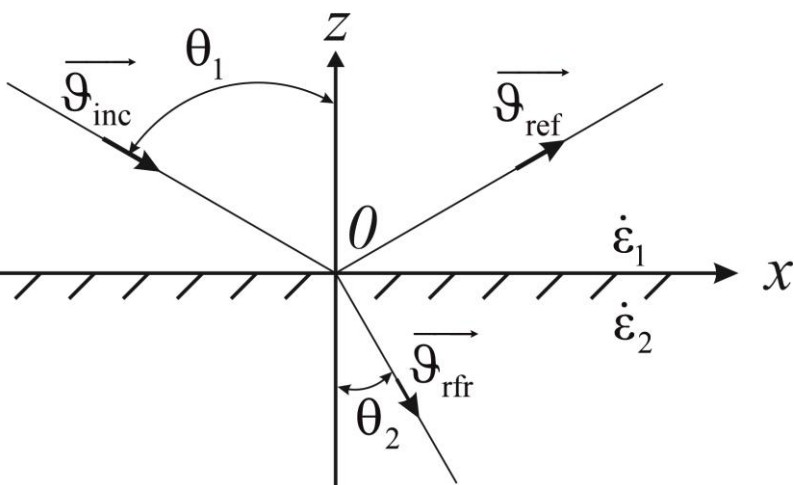

**Figure 2.** Geometry of reflection from a flat surface.

Although the Fresnel coefficients are generally defined for plane waves, if the irradiation takes place from a distance $H_0$ (which is significantly greater than the wavelength ($2\pi H_0 / \lambda \gg 1$)) and at least 4–5 Fresnel zones can be located within the reflecting surface, they are also valid for the specular reflection of spherical waves with fairly high accuracy. This means that the dimensions of the flat area of the reflective surface (where this model is valid) must be at least $3\sqrt{H_0 \lambda}$. For example, for a height of $H_0 = 1000$ m and $\lambda = 3$ cm, the diameter of the area is about 15 m.

In general, the dielectric constant is a complex number $\dot{\varepsilon} = \varepsilon - j60\lambda g$, where g is the medium's conductivity (reciprocal of resistivity), and $\lambda$ is a wavelength.

The most common case in practice is when the medium above the boundary is air ($\dot{\varepsilon}_1 \approx 1$ and $\dot{\varepsilon}_2 \approx \dot{\varepsilon}$); then, the Fresnel reflection coefficients are as follows:

$$\dot{K}_{f\ H} = \frac{\cos \theta_1 - \sqrt{\dot{\varepsilon} - \sin^2 \theta_1}}{\cos \theta_1 + \sqrt{\dot{\varepsilon} - \sin^2 \theta_1}}, \dot{K}_{f\ V} = \frac{\dot{\varepsilon} \cos \theta_1 - \sqrt{\dot{\varepsilon} - \sin^2 \theta_1}}{\dot{\varepsilon} \cos \theta_1 + \sqrt{\dot{\varepsilon} - \sin^2 \theta_1}}. \tag{5}$$

The radio brightness temperature of the thermal radiation of a flat surface is

$$T_{Br\ V(H)} = (1 - \left|\dot{K}_{f\ V(H)}\right|^2)T_0, \tag{6}$$

where $T_0$ is the thermodynamic surface temperature.

It should be noted that if the angle $\theta_1 = 0$, then in (5, 6), $\left|\dot{K}_{f\ V}\right|^2 = \left|\dot{K}_{f\ H}\right|^2$, and the polarization degree m = 0, i.e., at vertical sensing, the flat surface is a polarization–isotropic object—a source of unpolarized radio-thermal radiation. At Brewster's angle $\theta_B$ (determined by the equation $\sin \theta_B = \frac{1}{\sqrt{(1+\varepsilon)}}$), $\dot{K}_{fB} = 0$ and $\chi_B = 1$. The brightness temperature, in this case, is equal to the thermodynamic temperature, i.e., $T_{BrB} = T_0$.

If the medium under the boundary is non-isothermal (the medium parameters are not constant and depend on the direction), then [22,23,29]

$$T_{Br\ V(H)} = \left[1 - \left|\dot{K}_{f\ V(H)}(\theta)\right|^2\right]\chi(\theta) \int_0^\infty T_0(z)e^{-\chi(\theta)z}dz, \theta_1 = \theta, \tag{7}$$

where

$$\chi(\theta) = \frac{4\pi}{\lambda} \sqrt{\frac{\varepsilon\, r - \sin^2 \theta}{2} \left[ \sqrt{1 + \left( \frac{\varepsilon\, i}{\varepsilon\, r - \sin^2 \theta} \right)^2} - 1 \right]}$$

is the specific power attenuation of incident waves in the subsurface medium; $\varepsilon\, r, \varepsilon\, i$ are the real and imaginary parts of the complex dielectric constant.

We denote $K(h_0, \theta) = \exp\left\{ -\int_0^{h_0} \chi_A(z')dz' / \cos \theta \right\}$ (where $\chi_A$ is the attenuation in the atmosphere). Then, the flat surface temperature, considering the illumination by the atmosphere, can be written in the following form [22,23]:

$$T_{Br\, K\, V(H)} = \chi_{V(H)}(\dot{\varepsilon}, \theta)K(h_0, \theta)T_0 + \left| \dot{K}_{f\, V(H)}(\dot{\varepsilon}, \theta) \right|^2 K(h_0, \theta)T_{Br\, A}(\theta)$$
$$+ T_A[1 - K(h_0, \theta)], \tag{8}$$

where $\chi_{V(H)}(\varepsilon, \theta) = (1 - \left| \dot{K}_{f\, V(H)}(\varepsilon, \theta) \right|^2)$ is the emissivity of the surface; $T_{Br\, A}(\theta)$ is the radio brightness temperature of the total atmospheric radiation reflected by the surface in the $\theta$ direction; and $T_A$ is the average atmospheric temperature (approximately $30°$ less than the atmospheric temperature near the Earth).

$$T_{Br\, A}(\theta) = T_A \left[ 1 - e^{\frac{-1}{\cos \theta}(\chi_{oo}z_o + \chi_{wo}z_w)} \right], \tag{9}$$

where $z_o, z_w$ are the characteristic oxygen and water vapor absorption heights; $\chi_{oo}, \chi_{wo}$ are oxygen and water vapor absorption coefficients (near the Earth's surface).

In (8), the first term is the surface radiation in the $\theta$ direction, attenuated by the $K(h_0, \theta)$ factor on the wave propagation path from the surface to the receiving antenna; the second term is the total atmospheric radiation reflected by the surface in the $\theta$ direction and attenuated on the propagation path; and the third term is the direct atmospheric radiation between the receiving antenna and the surface in the $\theta$ direction.

In practice, the following expression is used to determine the coefficient $K(h_0, \theta)$:

$$K(h_0, \theta) = \exp\left\{ -\frac{1}{\cos \theta}(\chi_{oo}z_o + \chi_{wo}z_w) \right\}.$$

The above expressions, within the flat surface model, are the initial expressions for solving the inverse problems of data interpretation of the recorded reflected signals and signals of its radio-thermal radiation.

The numerical values of the real and imaginary parts of the complex dielectric constant used in practical calculations are given in Table 2 [22]. It is important to note that the values given in the table are only approximate. This is due to the relation between the dielectric constant and the system parameters (wavelength) and the surface parameters (e.g., salinity) [41–44].

**Table 2.** Electrophysical parameters of various media.

| Medium | Dielectric Constant $\varepsilon$ | Conductivity g, S/m |
|---|---|---|
| Snow | 1.2 | $2 \cdot 10^{-4}$ |
| Dry soil | 2.5...4 | $10^{-2} \dots 10^{-1}$ |
| Moist soil | 4...20 | $10^{-2} \dots 3$ |
| Crystalline rocks | 5...10 | $10^{-6} \dots 10^{-4}$ |
| Water | 60...80 | $10^{-3} \dots 10$ |

For the initial data (thermodynamic temperature $T_0 = 300$ K, the real part of the dielectric constant $\varepsilon = 70$, the conductivity of the medium g = 5 S/m, and the wavelength $\lambda = 3.333 \cdot 10^{-3}$ m (frequency f = 90 GHz)), the following dependencies, in accordance with the theoretical information given earlier (Figures 3–5), are obtained.

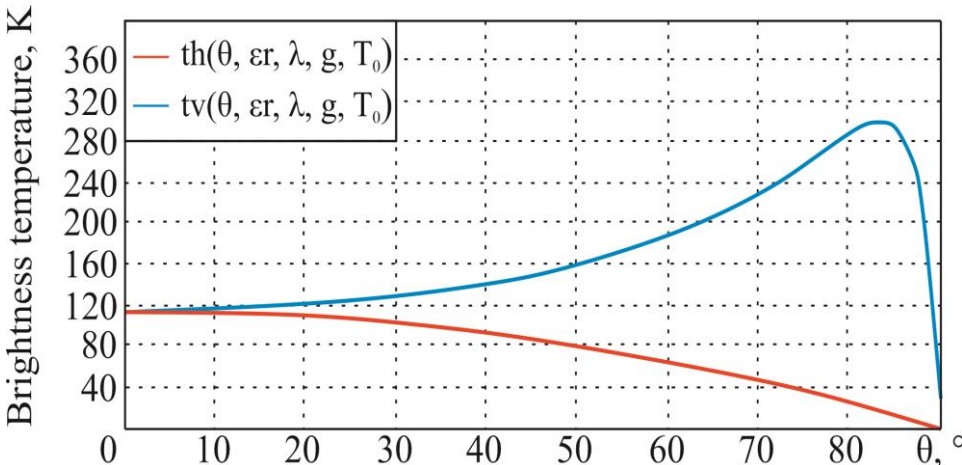

**Figure 3.** Dependencies of the flat surface brightness temperature on the sighting angle θ at horizontal th and vertical tv polarization.

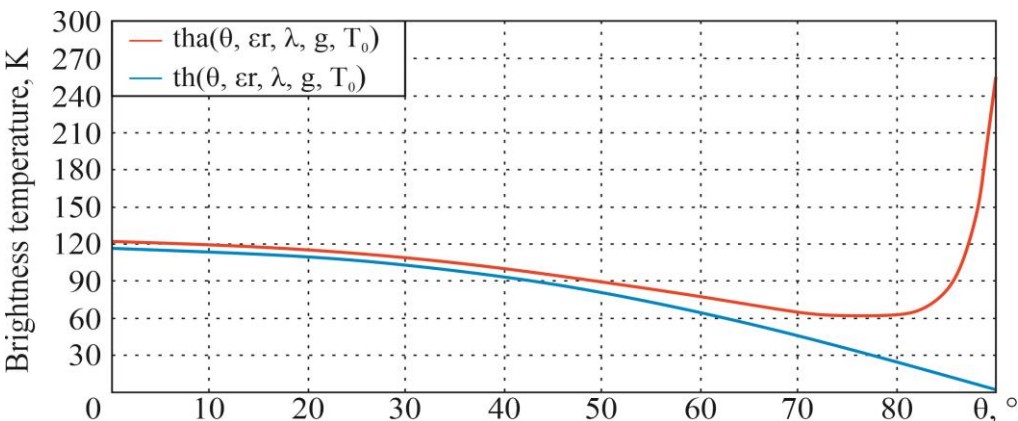

**Figure 4.** Dependencies of the flat surface brightness temperature on the sighting angle θ at horizontal polarization with tha and without th atmosphere impact.

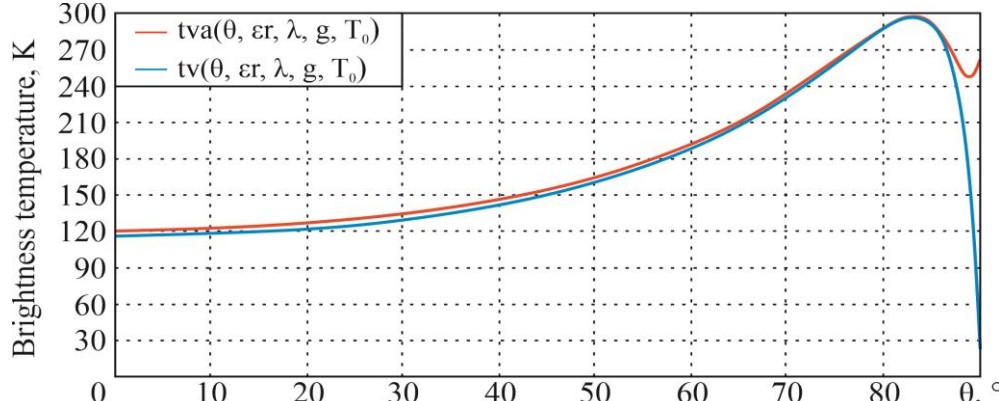

**Figure 5.** Dependencies of the flat surface brightness temperature on the sighting angle θ at vertical polarization with tva and without tv atmosphere impact.

3.1.2. Surface Model with Small-Scale Roughness

The small-scale model can be applied to rough surfaces where the roughness $h(x, y)$ is small in comparison to the electromagnetic wavelength, i.e., $|h(x, y)| \ll \lambda$, and the sloping $\dfrac{\partial h(x, y)}{\partial x} \ll 1, \dfrac{\partial h(x, y)}{\partial y} \ll 1$. The average value of the function $h(x, y)$ for a small and rough surface is the plane $h_0(x, y) = 0$. According to Rayleigh's criterion, this surface is almost smooth. The electrophysical characteristics of the medium under the surface $h(x, y)$ are as follows: dielectric constant $\varepsilon_2 \neq 1$ and permeability $\mu_2 = 1$ are constant values (the medium is isotropic). Such a model corresponds to the asphalt or concrete surface for the centimeter range of radio waves or an arable area for longer waves.

A field that is backscattered by such a surface (Figure 6) is in the small perturbation approximation [22,23]. The perturbed fields of different polarizations within the first approximation of small perturbations in media 1 and 2 are represented as follows:

$$\dot{E}_k = \dot{C}_0 \dot{M}_k(\dot{\varepsilon}_1, \dot{\varepsilon}_2, \vec{\vartheta}_i, \vec{\vartheta}_S) \cos \theta_i \cos \theta_S \frac{k^2}{\pi} \int\limits_{\Delta S} h(\vec{r}) e^{-jq_\perp(\vec{r} - \vec{r}_0)} d\vec{r},$$

where $\dot{C}_0 \cos \theta_i \cos \theta_S \int\limits_{\Delta S} h(\vec{r}) e^{-jq_\perp(\vec{r} - \vec{r}_0)} d\vec{r} = \dot{\alpha}$ is a coefficient that takes into account the transmission coefficient of the receiver path and losses in the atmosphere ($\Delta S$ is the surface area, and $h(\vec{r})$ is the height of roughness); $\vec{q}_\perp$ is the horizontal projection of the backscattering vector $\vec{q}$.

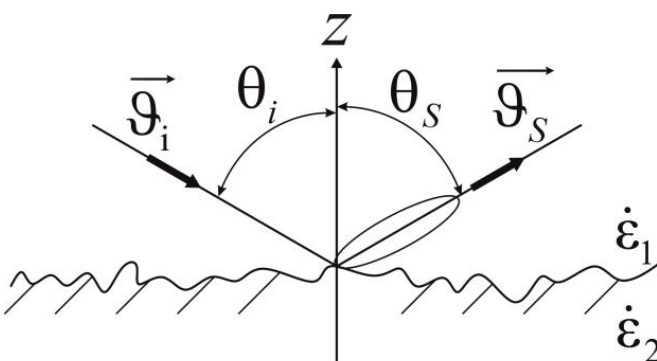

**Figure 6.** Geometry of backscattering by a surface with small-scale roughness.

The scattering coefficient $\dot{M}_k$ determines the relation between the parameter values of medium 1 and medium 2 and the received field voltages ($k = (VV, HH, HV, VH)$ indicates the type of polarization). These coefficients in the general case (for a bistatic model) were obtained in [22,23]. A special case of backscattering at a monostatic location is of particular importance $(\dot{\varepsilon}_1 = 1, \dot{\varepsilon}_2 = \dot{\varepsilon}, \cos \theta_i = \cos \theta_S, \phi_S = \pi)$. In this case,

$$\dot{M}_{HH} = \frac{\dot{\varepsilon} - 1}{\left(\cos \theta_i + \sqrt{\dot{\varepsilon} - \sin^2 \theta_i}\right)^2}, \dot{M}_{VV} = \frac{(\dot{\varepsilon} - 1) \cdot \left((\dot{\varepsilon} - 1) \sin^2 \theta_i + \dot{\varepsilon}\right)}{\left(\dot{\varepsilon} \cos \theta_i + \sqrt{\dot{\varepsilon} - \sin^2 \theta_i}\right)^2}, \tag{10}$$

$$\dot{M}_{VH} \approx \dot{M}_{HV} \approx 0.$$

Within the first approximation of small perturbations, the brightness temperature of the small-scale surface is

$$T_{Br\ V(H)} = \left(1 - K_{I\ V(H)}\right) T_0.$$

In the specific case of $l_h\left(\vec{r}\right)/\lambda >> 1$ (the indicatrix of the scattered field is narrow),

$$T_{Br\ V(H)} = \left(1 - \left|\dot{K}_{f\ V(H)}\right|^2 \cdot \exp\left\{-\left(\frac{4\pi}{\lambda} \cdot \sigma_h^2 \cdot \cos\theta\right)^2\right\} + K_{Dif\ V(H)}\right)T_0. \qquad (11)$$

The integral scattering coefficients that determine the increase in radio brightness temperatures can be represented as the sum of two terms, the first of which corresponds to the coherent component of the scattered field (unperturbed field $\vec{E}(x, y, z = 0)$), and the second to the diffuse component (the perturbed field).

$$K_{Coh\ V(\Gamma)} = \left|\dot{K}_{f\ V(H)}\right|^2 \cdot \exp\left\{-\left(\frac{4\pi}{\lambda} \cdot \sigma_h^2 \cdot \cos\theta\right)^2\right\}, \qquad (12)$$

$$K_{Dif\ V(H)} = \frac{1}{4\pi} \sum_{k=(V,H)} \int_\Omega K_{Dif\ V(H)}\left(\dot{\varepsilon}, \vec{\vartheta}_i, \vec{\vartheta}_S\right)d\Omega_S = \sigma_k^0\left(\dot{\varepsilon}, \vec{\vartheta}_i, \vec{\vartheta}_S\right)/\cos\theta_i. \qquad (13)$$

With a narrowly scattered field indicatrix (when the radius of roughness correlation is much larger than the wavelength),

$$K_{Dif\ k} \approx 16\pi k^2 \sigma_h^2 \cos^2\theta_i \left|\dot{M}_k\right|^2, \qquad (14)$$

where $k = \frac{2\pi}{\lambda}$ is the wavenumber.

The obtained relations relating the scattered fields and radio brightness of the surface temperature to its parameters are the basis for estimating these parameters. It should be noted that for more reliable estimations of these parameters (when calculating the radio brightness temperatures of a surface with small-scale roughness), the second approximation in the small perturbation method should be taken into account.

For the initial data (thermodynamic temperature $T_0 = 300$ K, real part of the dielectric constant $\varepsilon = 70$, conductivity of the medium $g = 5$ S/m, wavelength $\lambda = 3.333 \cdot 10^{-3}$ m (frequency $f = 90$ GHz), and root mean square of the roughness height $\sigma_h^2 = 10^{-4}$ m), the following dependencies, in accordance with the theoretical information given earlier, are obtained (Figure 7).

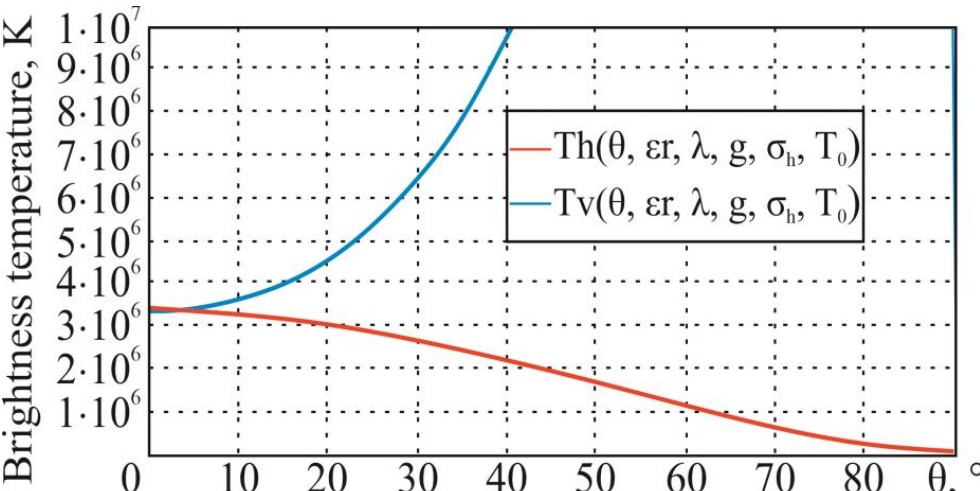

**Figure 7.** Dependences of the small-scale surface brightness temperature on the sighting angle $\theta$ at horizontal Th and vertical Tv polarization.

### 3.1.3. Surface Model with Large-Scale Roughness

Suppose that a surface with large-scale roughness satisfies the Kirchhoff approximation (consisting of large smooth roughness whose curvature radii are much larger than the wavelength, Figure 8). The field on the surface is defined as the sum of the incident and reflected waves [22,23]. In other words, the "large" scale of the roughness is determined by the curvature radii $R_K$, which must be much larger than the wavelength: $R_{Kx} \gg \lambda, R_{Ky} \gg \lambda$. There are no restrictions on the roughness height $h(x, y)$: it can be either significantly larger or smaller than the wavelength $\lambda$. The average height of the roughness is zero. In most cases, the reflective medium is isotropic, nonmagnetic ($\mu_2 = 1$), and described by a complex permittivity $\dot{\varepsilon}_2$. A typical illustration of this model is a "dead" ripple at sea.

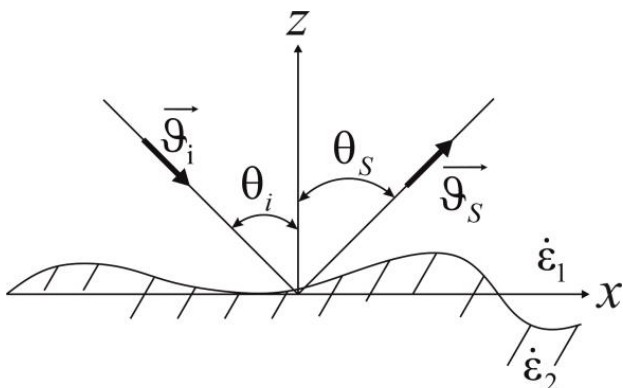

**Figure 8.** Surface with large-scale roughness.

In general, as we know, the brightness temperature is determined as follows:

$$T_{Br} = T_0 \chi,$$

where $\chi$ is the emissivity coefficient; $T_0$ is the thermodynamic temperature.

There are several approaches that we may take when attempting to calculate the brightness temperatures of this surface: the method without and with taking shading into account, based either on a deterministic averaging over the plane D of the emissivity $\chi_{V(H)}$ of quasi-flat elements of the large-scale roughness,

$$\chi_{V(H)} = \eta \left( \frac{\cos \theta_i}{\cos \theta_{i0}} \right) \left\{ 1 - \left| \dot{K}_{f\,V(H)}(\dot{\varepsilon}, \theta) \right|^2 \pm \sin^2 \alpha \left( \left| \dot{K}_{f\,V}(\dot{\varepsilon}, \theta) \right|^2 - \left| \dot{K}_{f\,H}(\dot{\varepsilon}, \theta) \right|^2 \right) \right\}, \quad (15)$$

with a periodic law of change in their height h, or on statistical averaging with given probabilities of element shading (probabilities of elements' visibility from specified directions in the geometric optics approximation) and the joint distribution probabilities of derivatives $\partial h/\partial x, \partial h/\partial y$. In this expression, the coefficient $\eta$ is equal to 1 or 0, depending on whether the area ds is visible or not from the antenna side; $\theta$ is the angle between the direction of the plane wave's wavevector k (direction towards the receiving antenna phase center) and the normal vector to the element ds; $\theta_{i0}$ is the angle between the direction k and the normal to the underlying surface; and $\alpha$ is the dihedral angle between the vertical plane of wave propagation and the plane formed by the vectors k and n.

For the initial data (the thermodynamic temperature $T_0 = 300$ K, real part of the dielectric constant $\varepsilon = 70$, conductivity of the medium g = 5 S/m, wavelength $\lambda = 3.333 \cdot 10^{-3}$ m (frequency f = 90 GHz), coefficient $\eta = 1$, and $\alpha = 45°$), the following dependencies, in accordance with the theoretical information given earlier, are obtained (Figure 9).

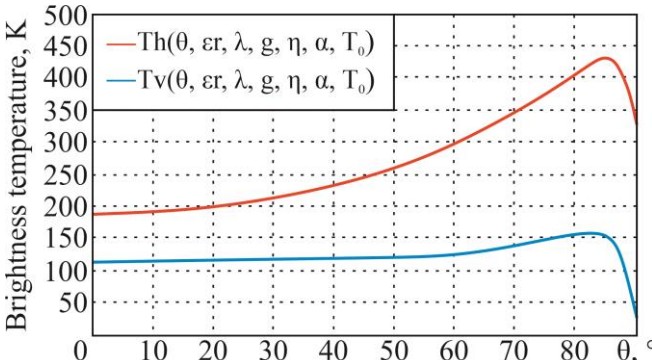

**Figure 9.** Dependences of the large-scale roughness surface brightness temperature on the sighting angle θ at horizontal Th and vertical Tv polarizations.

3.1.4. Two-Scale Surface Model

We consider a model that represents a combination of small smooth roughnesses that satisfy the conditions of small perturbations that cover the whole of the large quasi-flat roughness (Figure 10) [22,23,43–45].

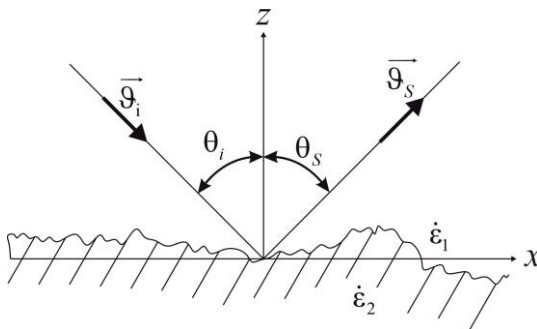

**Figure 10.** Two-scale surface.

This model is described by a wide spatial range of roughnesses, including examples of both small and large roughnesses. An example of such surfaces is the sea surface in a situation with heavy waves, when it is covered with both primary (large) waves and smaller waves (fine ripples) that are located on the surface of the large waves. Another example is a desert with sand barchans.

Nowadays, various versions of such a model have been developed and continuously improved, focusing on solving specific types of remote sensing applications of the disturbed sea surface using radar and radiometry methods. There are, for example, models that take backscattering into account in a quasi-mirror area, using the idea of a set of smooth platforms that are inclined at different angles with a local reflection coefficient. These local reflection coefficients are calculated using the Fresnel formula for a specific inclination angle of the selected platform to the local normal. In other models [44,45], "large" wave platforms are considered to be rough, and it is assumed that the specular component for the selected rough platform is reduced due to "resetting" by the diffuse scattering mechanism. For such a surface, assuming a normal spectral density of small-scale elevations with a height variance $\sigma_h^2$, which satisfies the condition of small perturbations, for the modified reflection coefficient on a large area $R_s(\lambda, \theta)$, taking into account this roughness, the following expression was obtained:

$$R_s(\lambda, \theta) = R(\lambda, \theta)\left(1 - 4 \cdot k^2 \cdot \sigma_h^2 \cdot \cos\theta \cdot \exp\left(-\frac{\sin^2\theta}{2}\right)\right),$$

where $k = \frac{2\pi}{\lambda}$ is the electromagnetic wavenumber.

In [43], the decrease in the Fresnel coefficients $R(\lambda, \theta)$, under the same conditions, is estimated to be of the order value of

$$\Delta(R(\lambda, \theta)) \simeq 2 \cdot k^2 \cdot \sigma_h^2 \cdot \cos^2 \theta.$$

For the initial data (thermodynamic temperature $T_0 = 300$ K, real part of the dielectric constant $\varepsilon = 70$, conductivity of the medium g $= 5$ S/m, and wavelength $\lambda = 0.03$ m (frequency f $= 10$ GHz)), the following dependencies, in accordance with the theoretical information given earlier, are obtained (Figures 11 and 12).

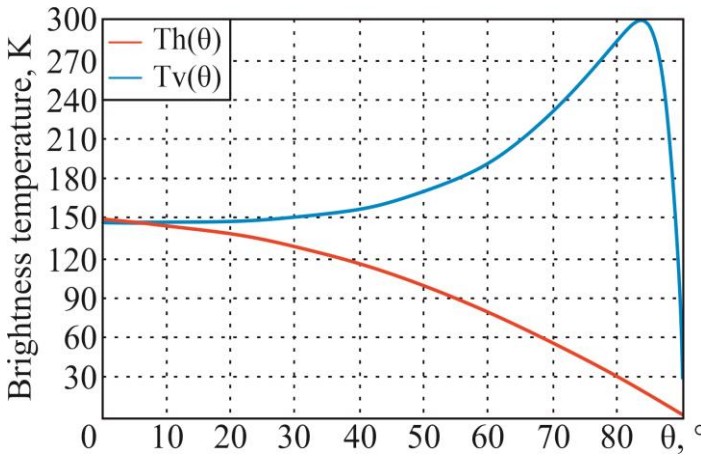

**Figure 11.** Dependences of the two-scale surface brightness temperature on the sighting angle θ at horizontal Th and vertical Tv polarizations.

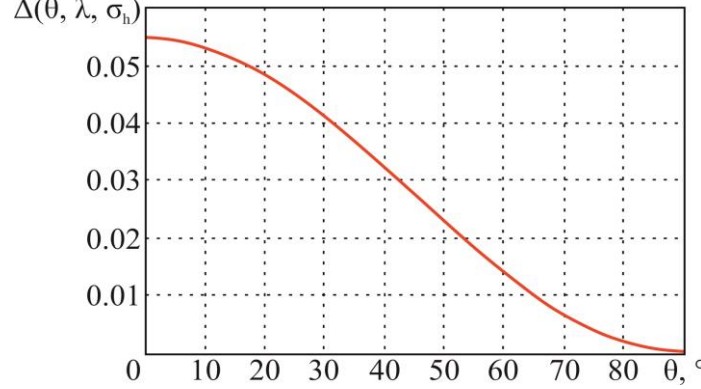

**Figure 12.** Dependence of the Fresnel coefficients decrease for two-scale surface on the sighting angle θ.

### 3.2. Empirical Models of Brightness Temperature

These models are obtained by statistical processing of data from multiple targeted experiments, as a result of which the correlation relations of averaged measured values and required parameters are established in the form of regression coefficients. Most often, regression equations are used, in which the recorded (averaged) and estimated values are defined either in the form of linear or nonlinear relationships or in the form of linear relationships and found correlations between the desired parameters and logarithms of the recorded values.

A large number of different useful models of the relationship between radio-thermal temperatures and the parameters of these media have been developed to interpret data from polarization measurements of radio-thermal radiation from land covers, water surfaces, "atmosphere-underlying surface" systems, "ocean–atmosphere" systems, and others. In

these systems, due to mutual illumination, the brightness temperatures are total and are often called apparent temperatures.

In the higher part of the centimeter wavelength range, the illumination by atmospheric radiation can be neglected, while in the lower part, the absorption and, consequently, its emissivity are noticeable.

The illumination leads, on the one hand, to errors in the estimations of the Earth surface parameters, while on the other hand (with the proper experiment organization), it leads to the possibility of measuring, along with the surface parameters (e.g., thermodynamic temperature $T_0$, near-water speed $v$ (m/s), etc.), the atmospheric parameters (atmospheric Q [kg/m$^2$] and cloud W [kg/m$^2$] moisture content, water storage R or intensity I [mm/h] of rains, etc.).

### 3.2.1. Sea Surface with Foam

The sea surface model with foam (excluding illumination) [24,25,28] is interesting, because the regression relations include the wind speed V as a parameter. It makes it possible to estimate this characteristic. The brightness temperature of the sea surface $T_{Br}$ is related to its thermodynamic temperature $T_0$ through the surface absorption coefficient k by a ratio:

$$T_{Br} = kT_0. \tag{16}$$

The absorption coefficient can be represented as

$$k = (k_0 + \Delta k_1)(1 - F) + \Delta k_2 F, \tag{17}$$

where $k_0$ is the absorption coefficient of a smooth sea surface. It can be calculated by the following formula:

$$k_{0p} = 1 - |R_p(\theta, \varepsilon)|^2, \tag{18}$$

where $R_p(\theta, \varepsilon)$ are the Fresnel coefficients for the p-th polarization; $\Delta k_1$ is the heavy sea correction; $\Delta k_2$ is the foam correction; and F is the effective surface area occupied by the foam.

These values for vertical and horizontal polarization are determined by the following regression relations:

$$\Delta k_{1h} = \frac{V}{T_0}\left(A + B\theta^2\right)\sqrt{f} - 0.00065f, \tag{19}$$

$$\Delta k_{1V} = \frac{V}{T_0}(a + b\exp(c\theta))\sqrt{f} - 0.00065f, \tag{20}$$

$$\Delta k_{2H} = 0.005 + \frac{208 + 1.29f}{T_0}\left(1 - 1.748 \cdot 10^{-3}\theta - 7.336 \cdot 10^{-5}\theta^2 + 1.044 \cdot 10^{-7}\theta^3\right),$$

$$\Delta k_{2V} = 0.005 + \frac{208 + 1.29f}{T_0}\left(1 - 9.946 \cdot 10^{-4}\theta + 3.218 \cdot 10^{-5}\theta^2 - 1.187 \cdot 10^{-6}\theta^3 + 7 \cdot 10^{-20}\theta^{10}\right),$$

$$F = b_0 + b_1 V + b_2 V^2. \tag{21}$$

The coefficients in expressions (17)–(21) have the following values:

$$A = 0.115, \ B = 3.8 \cdot 10^{-5}, \ a = 0.117, \ b = -2.09 \cdot 10^{-3}, \ c = 7.32 \cdot 10^{-2},$$

$$b_0 = 1.707 \cdot 10^{-2} + 8.56 \cdot 10^{-4}f + 1.12 \cdot 10^{-5}f^2,$$

$$b_1 = -1.501 \cdot 10^{-2} + 1.821 \cdot 10^{-3}f - 4.634 \cdot 10^{-5}f^2,$$

$$b_2 = 2.442 \cdot 10^{-4} - 2.282 \cdot 10^{-6}f + 4.134 \cdot 10^{-7}f^2.$$

The thermodynamic temperature $T_0$ is measured in Kelvin (K), the sighting angle $\theta$ in degrees, the wind speed V m/s, and the frequency f in GHz.

For the initial data (the thermodynamic temperature $T_0 = 300$ K, real part of the dielectric constant $\varepsilon = 70$, conductivity of the medium g = 5 S/m, wavelength $\lambda = 0.03$ m (frequency f = 10 GHz), and wind speed coefficient V = 4 m/s), the following dependencies, in accordance with the theoretical information given earlier, are obtained (Figure 13).

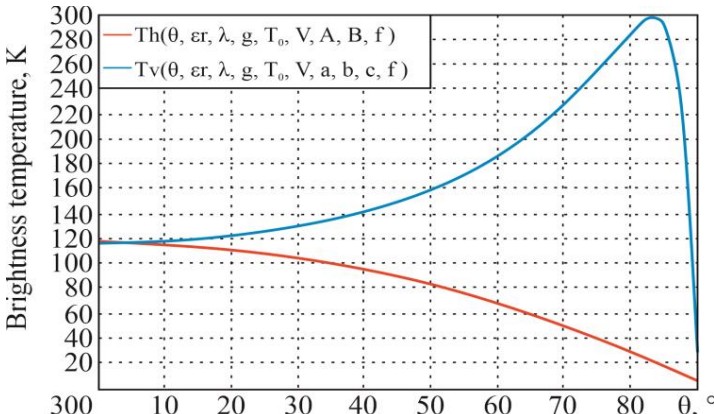

**Figure 13.** Dependences of the sea surface with foam brightness temperature on the sighting angle θ at horizontal Th and vertical Tv polarizations.

### 3.2.2. $\tau - \omega$ Model

The $\tau - \omega$ model [26,36] was developed based on the assumption that vegetation is a uniformly absorbing and backscattering layer above the soil surface. According to this model, the brightness temperature at the p-th polarization consists of the direct vegetation emission, soil emission, and vegetation emission reflected by the soil, and is described by the following equation:

$$
\begin{aligned}
T_{Br\,p} = \left(1 - R_p\right) \cdot T_s \cdot \exp\left(-\frac{\tau}{\cos\theta}\right) + (1 - \omega) \cdot T_v \cdot \left(1 - \exp\left(-\frac{\tau}{\cos\theta}\right)\right) + \\
R_p \cdot (1 - \omega) \cdot T_v \cdot \left(1 - \exp\left(-\frac{\tau}{\cos\theta}\right)\right) \cdot \exp\left(-\frac{\tau}{\cos\theta}\right),
\end{aligned}
\tag{22}
$$

where $R_p$ is the Fresnel coefficient at the p-th polarization; $T_s$ is the soil temperature, which is by assumption equal to the vegetation temperature $T_v$ due to the fact that the soil and vegetation are in temperature equilibrium; $\omega$ is the single backscattering coefficient (in the C-band (from 4 GHz to 8.8 GHz), it is equal from 0.05 to 0.13); and $\tau$ is the optical depth of vegetation.

For the initial data (the temperature of soil and vegetation $T_s = T_v = 293$ K, real part of the dielectric constant $\varepsilon = 10$, wavelength $\lambda = 0.06$ m (frequency f = 5 GHz), single backscattering coefficient $\omega = 0.09$, and optical depth of vegetation $\tau = 2$), the following dependencies, in accordance with the theoretical information given earlier, are obtained (Figure 14).

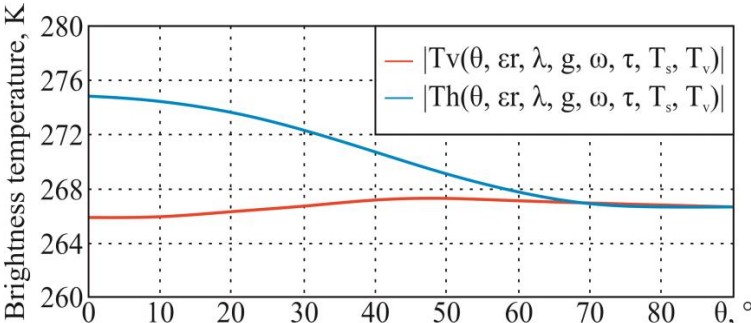

**Figure 14.** Dependences of the $\tau - \omega$ model brightness temperature on the sighting angle θ at horizontal Th and vertical Tv polarizations.

### 3.2.3. $Q_p$ Model

The $Q_p$ model can be used at large sighting angles for radiometers located on the Aqua (AMSR-E), Nimbus-7 (SSMR), and TRRM (TMI) satellites, and the DMSP (SSM/I) meteorological satellite. It is developed based on an advanced integral equation model for a wide range of surface moisture and roughness characteristics. For the $Q_p$ model, the effective reflectivity at the p-th polarization is determined by the backscattering coefficient [27]:

$$R_p^e = Q_p \cdot R_P + \left(1 - Q_p\right) \cdot R_P, \tag{23}$$

where $R_p$ is the Fresnel coefficient at the p-th polarization, and $Q_p$ is the roughness coefficient, which is expressed by

$$\log(Q_v) = 3.2165 + 2.4528 \cdot \log(s/l) - 6.6741 \cdot \log(s/l),$$
$$\log(Q_h) = 5.6036 + 3.0650 \cdot \log(s/l) - 9.3776 \cdot \log(s/l), \tag{24}$$

where s is the root mean square of the roughness height, and l is the correlation radius.

The brightness temperature is related to the backscattering coefficient by the following expression:

$$T_{Br\ p} = \left(1 - \left|R_p^e\right|^2\right) \cdot T_0, \tag{25}$$

where $T_0$ is the thermodynamic surface temperature.

For the initial data (thermodynamic temperature $T_0 = 300$ K, real part of the dielectric constant $\varepsilon = 5$, conductivity of the medium g = 1 S/m, wavelength $\lambda = 0.015$ m (frequency f = 20 GHz), root mean square of the roughness height s = 0.01, and correlation radius of roughness l = 0.05), the following dependencies, in accordance with the theoretical information given earlier, are obtained (Figure 15).

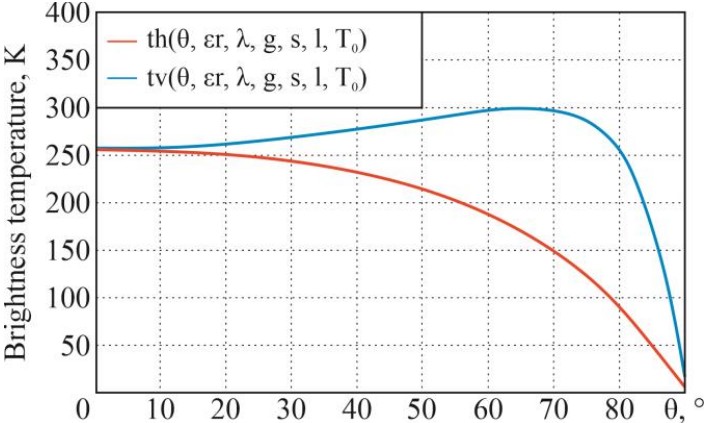

**Figure 15.** Dependences of the $Q_p$ model brightness temperature on the sighting angle θ at horizontal th and vertical tv polarizations.

### 3.2.4. Atmosphere–Surface Regression Model

In multi-parameter measurements in passive remote sensing systems, it is possible to estimate not only the underlying surface parameters, but the atmosphere layer between this surface and the receiving antenna. One of the estimated parameters can be the moisture content of the cloudless atmosphere Q [g/cm$^2$] (the integral water vapor content in the atmosphere volume of a unit's cross-section). When using the atmosphere-underlying surface regression model and space sensing, the attenuation coefficient on the wave propagation path from the surface to the receiving antenna can be considered equal to [28]

$$K(h_0, \theta) = K(h_0 = \infty, \theta) = \exp\left(\frac{-\Gamma}{\cos\theta}\right), \tag{26}$$

where $\Gamma$ is the attenuation, which is determined by the regression equation in relation to the moisture content Q:

$$\Gamma = \begin{cases} 0.06 + 0.014 \cdot Q \text{on } \lambda = 0.8 \text{ cm,} \\ 0.015 + 0.08 \cdot Q \text{on } \lambda = 1.35 \text{ cm,} \end{cases} \tag{27}$$

For the moisture content of $Q = 10$ [g/cm$^2$], the following dependencies, in accordance with the theoretical information given earlier, are obtained (Figure 16).

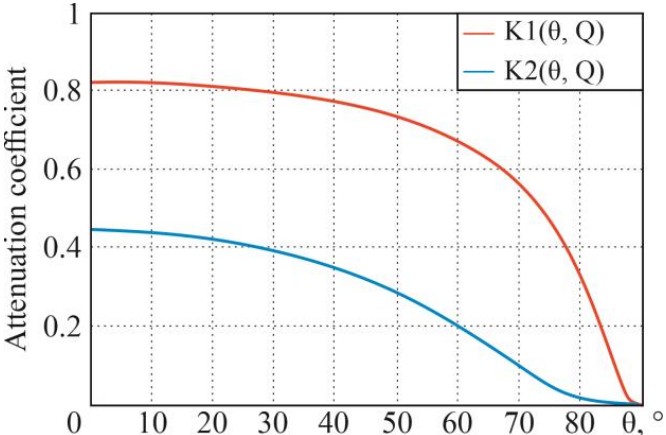

**Figure 16.** Dependence of attenuation coefficients on the sighting angle $\theta$, K1 ($\lambda = 0.8$ cm), and K2 ($\lambda = 1.35$ cm).

The regression relations for determining the atmosphere's moisture content Q and cloud moisture content W, obtained using the Meteor satellite equipment, at a sighting angle $\theta = 30°$ and an operating wavelength $\lambda = 0.8$ cm, have the following form [28]:

$$Q = 0.16 \cdot T_{Br\,H} - 23, W = 0.018 \cdot T_{Br\,H} - 2.78, \tag{28}$$

where $T_{Br\,H}$ is the brightness temperature of the surface at horizontal polarization.

According to the studies based on the Nimbus-5 (artificial Earth satellite) equipment, the following dependencies were obtained for nadir observations (sighting angle $\theta = 0°$) [28]:

$$Q = -4.03 \cdot T_{Br}(\lambda_2) - 0.0515 \cdot T_{Br}(\lambda_1),$$
$$W = -0.404 - 1.54 \cdot 10^{-3} \cdot T_{Br}(\lambda_2) + 4.09 \cdot 10^{-3} \cdot T_{Br}(\lambda_1), \tag{29}$$

where $\lambda_1 = 0.96$ cm and $\lambda_2 = 1.35$ cm are the operating wavelengths.

For the multichannel microwave radiometer of the "Seasat" satellite expressions for determining the near-surface speed $v_{ns} < 7$ m/s, the thermodynamic temperature $T_0$ [K], atmosphere's moisture content Q [g/cm$^2$], and cloud moisture content W [mg/cm$^2$] were obtained [28,40]:

$$v_{ns} = -523.9 + 0.2229 \cdot T_{Br\,V}(\lambda_4) + 0.6056 \cdot T_{Br\,H}(\lambda_4) + 130.3 \cdot \ln(280 - T_{Br\,V}(\lambda_3))$$
$$-39.19 \cdot \ln(280 - T_{Br\,H}(\lambda_3)) + 10.24 \cdot \ln(280 - T_{Br\,V}(\lambda_2)) - 32.75 \cdot \ln(280 - T_{Br\,H}(\lambda_2)) \tag{30}$$
$$+2.999 \cdot \theta,$$

$$T_0 = -149.1 + 1.677 \cdot T_{Br\,V}(\lambda_5) + 1.666 \cdot T_{Br\,H}(\lambda_5) - 0.2767 \cdot T_{Br\,V}(\lambda_4) - 0.559 \cdot T_{Br\,H}(\lambda_4)$$
$$+46.17 \cdot \ln(280 - T_{Br\,V}(\lambda_3)) + 3.097 \cdot \ln(280 - T_{Br\,H}(\lambda_3)) - 0.916 \cdot \ln(280 - T_{Br\,V}(\lambda_2)) \tag{31}$$
$$-12.54 \cdot \ln(280 - T_{Br\,H}(\lambda_2)) - 0.585 \cdot \theta,$$

$$W = 246.1 - 51.72 \cdot \ln(280 - T_{Br\,V}(\lambda_3)) + 134.4 \cdot \ln(280 - T_\Gamma(\lambda_3)) + 46,14 \cdot \ln(280 - T_{Br\,V}(\lambda_2))$$
$$+24.95 \cdot \ln(280 - T_{Br\,H}(\lambda_2)) - 155.5 \cdot \ln(280 - T_{Br\,V}(\lambda_1)) - 36.63 \cdot \ln(280 - T_{Br\,H}(\lambda_1)) \tag{32}$$
$$-3.391 \cdot \theta,$$

$$Q = -9.784 + 6.927 \cdot \ln(280 - T_{Br\,V}(\lambda_3)) + 5.361 \cdot \ln(280 - T_{Br\,H}(\lambda_3)) - 4.518 \cdot \ln(280 - T_{Br\,V}(\lambda_2))$$
$$-6.081 \cdot \ln(280 - T_{Br\,H}(\lambda_2)) + 0.039 \cdot \theta, \tag{33}$$

where $\lambda_1 = 0.81$ cm, $\lambda_2 = 1.43$ cm, $\lambda_3 = 1.67$ cm, $\lambda_4 = 2.8$ cm, and $\lambda_5 = 4.55$ cm are the operating wavelengths.

For the initial data (thermodynamic temperature $T_0 = 300$ K, real part of the dielectric constant $\varepsilon = 70$, conductivity of the medium $g = 5$ S/m), the following dependencies, in accordance with the theoretical information given earlier, are obtained (Figure 17).

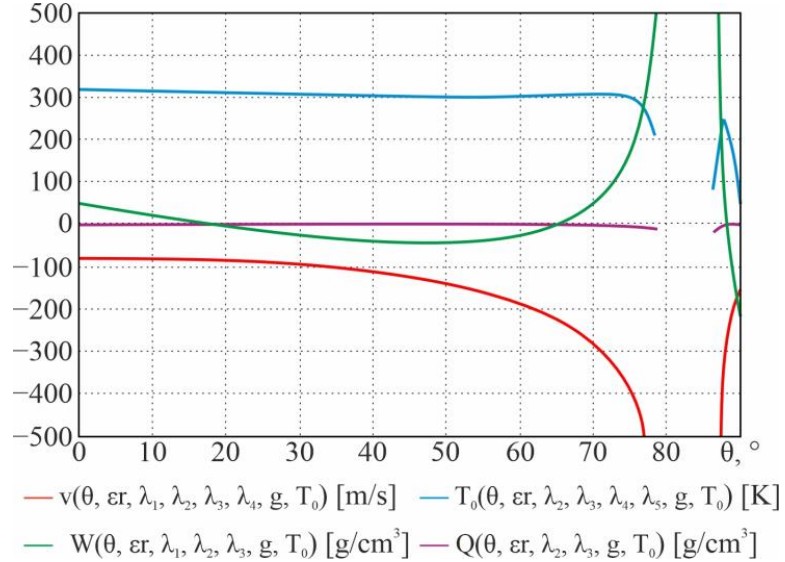

**Figure 17.** Dependences of the near-surface speed v, thermodynamic temperature $T_0$, atmosphere's moisture content Q, and cloud moisture content W on the sighting angle θ.

### 3.3. Electrodynamic Surface Models for Active Remote Sensing

Active sensing is based on the fact that radio engineering systems have a radiation source that is directed at the target object, and then, the reflected radiation is detected by the receiver [22,23].

One of the most important backscattering characteristics of an object (surface) is the radar cross-section [8,43].

$$\sigma = \frac{\Pi_A}{\Pi_T} 4\pi R^2, \tag{34}$$

where $\Pi_T$ and $\Pi_A$ are the power flux densities [W/m$^2$] of the incident electromagnetic wave (EMW) near the target and reflected near the antenna phase center; R is the distance from the antenna to the object.

In (34),

$$\prod_T = \frac{P_r G_r\left(\vec{\vartheta}_i\right)}{4\pi R_i^2} = \frac{\left|\dot{E}_{inc}\right|^2}{2\rho} = \frac{\left|\dot{E}_{inc}\right|^2}{240\pi},$$

$$\prod_A = \frac{P_{scat(refl)}}{4\pi R_S^2} = \frac{\prod_T \cdot \sigma}{4\pi R_S^2} = \frac{\left\langle \left| \dot{E}_{scat(refl)} \right|^2 \right\rangle}{2\rho} = \frac{1}{240\pi} \left\langle \left| \dot{E}_{scat(refl)} \right|^2 \right\rangle,$$

where $P_r$ is the radiation source's power, $G_r\left(\vec{\vartheta}_i\right)$ is the power gain (radiation pattern), $R_S$ is the distance between the field receiving point and the target, $R_i$ is the distance between the transmitting antenna and the target, $\dot{E}_{inc}$ is the complex amplitude of the incident field near the target area, $\dot{E}_{scat(refl)}$ is the complex amplitude of the scattered field strength at the receiving point, $P_{scat(refl)}$ is the power of the field, scattered (reflected) by the surface near it, and $\rho = \sqrt{\varepsilon_a / \mu_a}$ is the wave impedance of the wave propagation medium. For free space and approximately for the Earth's atmosphere, it is equal to $120\pi$ Ohm.

The power flux density in the antenna location area, assuming it is isotropically distributed on the sphere, is as follows:

$$\prod_A = \frac{P_{scat(refl)}}{4\pi R_S^2} = \frac{\prod_T \cdot \sigma}{4\pi R_S^2}.$$

The complex amplitude of the scattered field strength at the receiving point contains information about the surface from which the incident electromagnetic wave is reflected.

Each surface has individual geometric (height and flatness of roughnesses), electro-chemical (conductivity), and other features. Thus, the information about them is contained in the RCS, which helps to solve, for example, the inverse problem of the restoration (estimation) of surface parameters.

An important characteristic of the unit surface's backscattering (1 m$^2$) is the normalized radar cross-section (normalized RCS):

$$\sigma^o(x, y) = \frac{\Delta\sigma(x, y)}{\Delta x \Delta y}, \tag{35}$$

which for real land covers would be a function of the underlying surface coordinates (flat or spherical, passing at the midpoint of its roughness).

Using the expression for the specific RCS, and according to it, experimental studies and the development of empirical models can be performed.

$$\sigma^0 = 4\pi R_S^2 \frac{\prod_A}{\prod_A D_T} = \frac{(4\pi)^3}{\lambda^2} \frac{P_{k(rec)}}{P_{trans}} \frac{R_i^2 R_S^2}{G_{trans}(\vartheta_i) G_{rec}(\vartheta_S) D_T},$$

where $D_T$ is the target area; $P_{trans}$ and $P_{k(rec)}$ are the transmitter power and receiver output power; and $G_{trans}(\vartheta_i)$ and $G_{rec}(\vartheta_S)$ are the radiation patterns of the radiating and receiving antennas, located at a distance $R_i$ and $R_s$ from the target, respectively.

Otherwise,

$$P_k = \frac{\lambda^2}{(4\pi)^3} \frac{P_{trans} G_{trans}(\vec{\vartheta}_i) G_{rec}(\vec{\vartheta}_S) \sigma_k^0(\vec{\vartheta}_i, \vec{\vartheta}_S) D_T}{R_i^2 R_S^2}.$$

### 3.3.1. Electrodynamic Model of a Flat Surface

In active remote sensing of a flat surface, the complex amplitudes for vertical and horizontal polarization are equal [22,23].

$$\dot{A}_{VV} = \dot{\alpha} \cdot K_{f\,V}(\theta, \dot{\varepsilon}), \quad \dot{A}_{HH} = \dot{\alpha} \cdot K_{f\,H}(\theta, \dot{\varepsilon}), \tag{36}$$

where $\dot{\alpha}$ is a complex factor, which is equal for the complex amplitudes $\dot{A}_{VV}$ and $\dot{A}_{HH}$; $K_{fV}(\theta, \dot{\varepsilon})$ are the Fresnel reflection coefficients.

For the initial data (the real part of the dielectric constant $\varepsilon = 70$, conductivity of the medium g = 5 S/m, wavelength $\lambda = 0.03$ m (frequency f = 10 GHz), coefficient $\dot{\alpha} = 1 - j0, 1$), the following dependencies, in accordance with the theoretical information given earlier, are obtained (Figure 18).

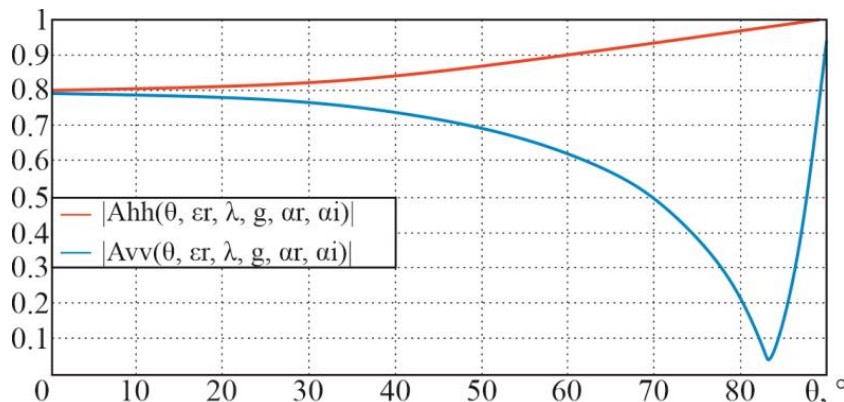

**Figure 18.** Dependences of complex amplitude moduli on the sighting angle θ at horizontal Ahh and vertical Avv polarizations.

### 3.3.2. Surface Model with Small-Scale Roughness

The normalized RCS of a small-scale roughness surface is determined by the following expression [22,23]:

$$\sigma_k^0 = \frac{4k^4}{\pi}\left|\dot{M}_k\right|^2 \cos^2\theta_i \cos^2\theta_S W\left[\vec{q}_\perp(\vec{r})\right], \tag{37}$$

where

$$W\left[\vec{q}_\perp(\vec{r})\right] = \int\limits_{-\infty}^{+\infty} R_h\left(\vec{r}, \Delta\vec{r}\right)\exp\left\{-j\vec{q}_\perp\Delta\vec{r}\right\}d\Delta\vec{r}, \tag{38}$$

$$R_h\left(\vec{r}, \Delta\vec{r}\right) = \left\langle h(\vec{r})h\left(\vec{r} + \Delta\vec{r}\right)\right\rangle \tag{39}$$

are the energy spectrum and correlation function, respectively.

In practice, the following expressions can be used to calculate the normalized RCS of this type of surface:

$$\sigma_{VV}^0 = 4\left(\frac{2\pi}{\lambda}\right)^4 \sigma_h^2 l_h^2 \left|\dot{M}_{VV}\right|^2 \cos^4\theta_i \exp\left[-\left(\frac{2\pi l_h}{\lambda}\right)\sin^2\theta_i\right], \tag{40}$$

$$\sigma_{HH}^0 = 4\left(\frac{2\pi}{\lambda}\right)^4 \sigma_h^2 l_h^2 \left|\dot{M}_{HH}\right|^2 \cos^4\theta_i \exp\left[-\left(\frac{2\pi l_h}{\lambda}\right)\sin^2\theta_i\right], \tag{41}$$

where $\lambda$ is the wavelength; $\sigma_h^2$ is the root mean square of the roughness height; $l_h$ is the distance (radius) of spatial correlation, which may be greater or less than the wavelength, assuming that the roughness is sloping; and $\theta_i$ is the sighting angle.

For the initial data (real part of the dielectric constant $\varepsilon = 70$, conductivity of the medium g = 5 S/m, wavelength $\lambda = 0.03$ m (frequency f = 10 GHz), root mean square of the roughness height $\sigma_h^2 = 0.001$, and roughness correlation radius $l_h = 0.001$), the following dependencies, in accordance with the theoretical information given earlier, are obtained (Figure 19).

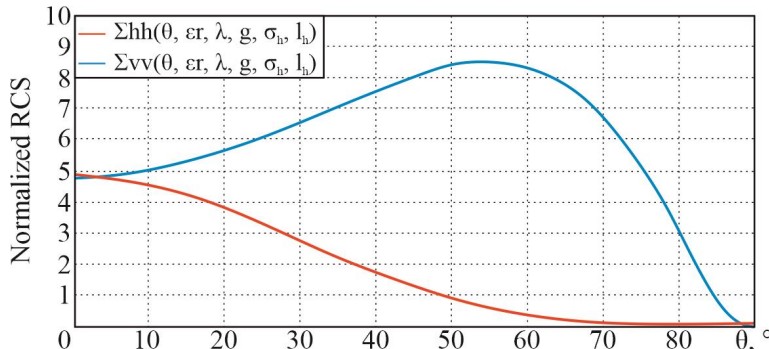

**Figure 19.** Dependences of small-scale surface normalized RCSs on the sighting angle θ at horizontal Σhh and vertical Σvv polarizations.

### 3.3.3. Surface Model with Large-Scale Roughness

The normalized RCS of such a surface with significantly sloping roughness (the depolarized component can be neglected) and at $\theta_i = \theta_S$, $\phi_S = \pi$ does not depend on the type of polarization [22,23,29].

$$
\sigma^o = \frac{\left|\dot{K}_f(0)\right|^2}{4\cos^4\theta_i}\frac{\updownarrow_h^2}{\sigma_h^2}\exp\left(-\frac{\updownarrow_h^2}{4\sigma_h^2}\mathrm{tg}^2\theta_i\right),
\tag{42}
$$

where $\updownarrow_h$ is the roughness correlation radius; $\sigma_h^2$ is the root mean square of the roughness height; $\theta_i$ s the direction of the electromagnetic wave incidence; and $\dot{K}_f(0) = \dot{K}_{f\,V}(\dot{\varepsilon}, \theta_i = 0) = \dot{K}_{f\,H}(\dot{\varepsilon}, \theta_i = 0)$.

For the initial data (real part of the dielectric constant $\varepsilon = 70$, conductivity of the medium g = 5 S/m, wavelength $\lambda = 0.03$ m (frequency f = 10 GHz), root mean square of the roughness height $\sigma_h^2 = 0.3$, and roughness correlation radius $l_h = 0.3$), the following dependencies, in accordance with the theoretical information given earlier, are obtained (Figure 20).

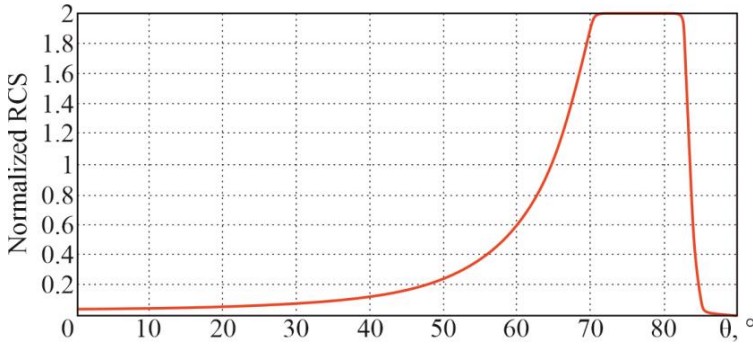

**Figure 20.** Dependences of a large-scale surface normalized RCS on the sighting angle θ.

### 3.3.4. Two-Scale Surface Model

The normalized RCS in the case of backscattering can be determined [22,23].

$$
\begin{aligned}
\sigma^o_{VV} &= \sigma^o_1 + \sigma^o_2\left|\dot{M}_{VV}(\dot{\varepsilon}, \theta_i)\right|^2, \\
\sigma^o_{HH} &= \sigma^o_1 + \sigma^o_2\left|\dot{M}_{HH}(\dot{\varepsilon}, \theta_i)\right|^2, \\
\sigma^o_{VH} = \sigma^o_{HV} &= \sigma^o_1\left[\begin{array}{c}\left|\dot{M}_{VV}(\dot{\varepsilon}, \theta_i)\right|^2 + \left|\dot{M}_{HH}(\dot{\varepsilon}, \theta_i)\right|^2 - \\ -2\left|\dot{M}_{VV}(\dot{\varepsilon}, \theta_i)\right|\left|\dot{M}_{HH}(\dot{\varepsilon}, \theta_i)\right|\cos\Delta\varphi\end{array}\right]\sigma^2_{h'1'},
\end{aligned}
\tag{43}
$$

where

$$\sigma_1^o = \frac{\left|\dot{K}_f(0)\right|^2}{a_{\text{ш}1}^2} \exp\left[-\frac{tg^2\theta_i}{a_{\text{ш}1}^2} - \left(\frac{4\pi\sigma_{h1}^2}{\lambda}\right)^2\right] \sec^4\theta_i,$$

$$\sigma_2^o = 4k^4\sigma_{h_2}^2 \updownarrow_{h_2}^2 \cos^4\theta_i \exp\left[-\left(\frac{2\pi\updownarrow_{h_2}}{\lambda}\right)^2 \sin^2\theta_i\right], \qquad (44)$$

where $a_{\text{ш}}^2 = \frac{4\sigma_h^2}{\updownarrow_h^2}$; $\Delta\varphi = \arg\left(\dot{M}_{VV}(\dot{\varepsilon}, \theta_i)\right) - \arg\left(\dot{M}_{HH}(\dot{\varepsilon}, \theta_i)\right)$, $\sigma_{h'1}^2 = 2\sigma_{h1}^2/l_{h1}^2$.

A distinctive feature of this model is the presence of the backscattering matrix element $\sigma_{VH}^o = \sigma_{HV}^o$, which describes the depolarization of the reflected signal. The variance of this element is proportional to the variance of the inclinations of the large-scale roughness surface.

For the initial data (real part of the dielectric constant $\varepsilon = 70$, conductivity of the medium g = 5 S/m, wavelength $\lambda = 0.03$ m (frequency f = 10 GHz), root mean square of the roughness height $\sigma_{h1}^2 = 0.3$, and roughness correlation radius of a major component $l_{h1} = 0.3$ (for a minor component, $\sigma_{h2}^2 = 0.001$, $l_{h2} = 0.001$)), the following dependencies, in accordance with the theoretical information given earlier, are obtained (Figure 21).

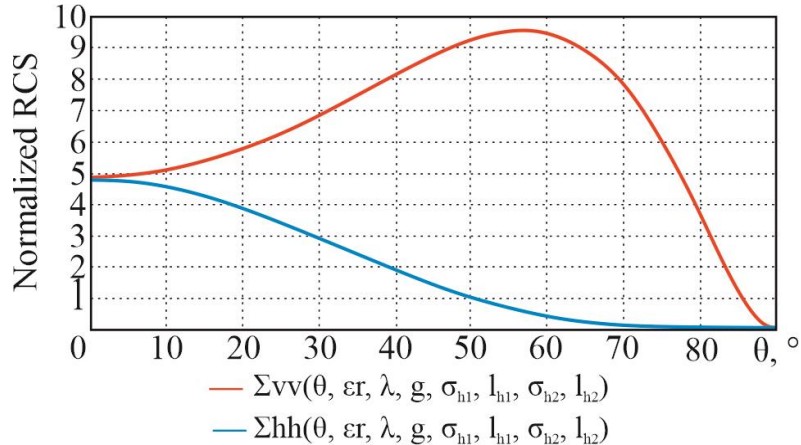

**Figure 21.** Dependences of a two-scale surface normalized RCS on the sighting angle $\theta$ at horizontal $\Sigma$hh and vertical $\Sigma$vv polarizations.

### 3.4. Empirical Models of Surfaces in Active Remote Sensing
3.4.1. Exponential Model

Let us consider the exponential model [31,32], which can be used to describe the RCSs of various surfaces (including quasi-smooth, rough with and without vegetation, as well as snow and anthropogenic areas) in the range of operating frequencies 3...100 GHz and at grazing angles $\psi \le 30°$.

When dealing with a beam that is nearly parallel to the surface, it is sometimes more useful to refer to the angle between the beam and the surface rather than the angle between the beam and the normal, i.e., an angle of 90° minus the angle of incidence. This is called the glancing angle or grazing angle. An incidence at a grazing angle is called a "grazing incidence". The grazing angle is the angle formed by an incident beam (or reflected beam) and a plane (surface).

The normalized RCS of such a model (in dB) is determined by

$$\sigma^0(f, \psi) = A_1 + A_2 \cdot \log\left(\frac{\psi}{20}\right) + A_3 \cdot \log\left(\frac{f}{10}\right), \qquad (45)$$

where $\psi$ is the grazing angle, and $\psi = \pi/2 - \theta$, $\theta$ is the angle of incidence; f is the operating frequency; and $A_1, A_2, A_3$ are coefficients that are determined by the type of surface (Table 3).

**Table 3.** Values of coefficients A1, A2, and A3 in Formula (45).

| Surface | $A_1$ | $A_2$ | $A_3$ |
|---|---|---|---|
| Concrete | −49 | 32 | 20 |
| Arable land | −37 | 18 | 15 |
| Snow | −34 | 25 | 15 |
| Deciduous forest, summer | −20 | 10 | 6 |
| Deciduous forest, winter | −40 | 10 | 6 |
| Coniferous forest, summer and winter | −20 | 10 | 6 |
| Meadow, grass height over 0.5 m | −21 | 10 | 6 |
| Meadow, grass height less than 0.5 m | −28 | 10 | 6 |
| Urban and rural buildings | −8.5 | 5 | 3 |

For the operating frequency f = 50 GHz, the following dependencies, in accordance with the theoretical information given earlier, are obtained (Figure 22).

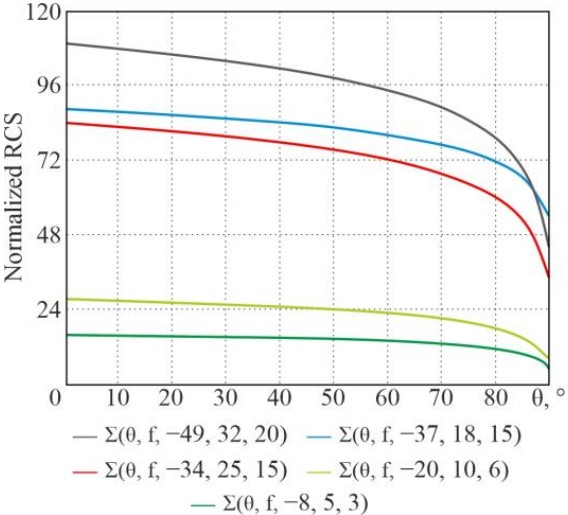

**Figure 22.** Dependences of the exponential model's normalized RCS on the sighting angle θ for different surface types.

3.4.2. Oh's Model

Consider the empirical Oh's model proposed in [33,34]. This model is based on radar backscattering measurements and information about the scattering pattern in the boundary cases (when the parameter characterizing the height of the roughness $k \cdot \sigma_h$ is large). Here, $\sigma_h$ is the root mean square height of the roughness, and k is the wavenumber. The proposed model operates under the following conditions: roughness characteristics $0.1 < k \cdot \sigma_h < 6.0$, $2.6 < k \cdot \updownarrow < 19.7$ ($\updownarrow$ is the roughness correlation radius); and the moisture content is $0.09 < m < 0.31$. The expressions for the normalized RCSs at the vertical-, horizontal-, and cross-polarizations for this model are as follows:

$$\sigma_{VV}^0 = \frac{g}{\sqrt{p}} \cdot \cos^3 \theta \cdot (\Gamma_V(\theta) + \Gamma_H(\theta)),$$
$$\sigma_{HH}^0 = g \cdot \sqrt{p} \cdot \cos^3 \theta \cdot (\Gamma_V(\theta) + \Gamma_H(\theta)), \tag{46}$$
$$\sigma_{HV}^0 = q \cdot \sigma_{VV}^0,$$

where $\Gamma_V(\theta), \Gamma_H(\theta)$ are the Fresnel coefficients at vertical and horizontal polarizations;

$$g = 0.7 \cdot \left(1 - \exp\left(-0.65 \cdot (k \cdot \sigma_h)^{1.8}\right)\right),$$

$$p = \frac{\sigma_{HH}^0}{\sigma_{VV}^0}, q = \frac{\sigma_{HV}^0}{\sigma_{VV}^0},$$

$$p = \left(1 - \left(\frac{2\theta}{\pi}\right)^{1/3\Gamma_0} \cdot \exp(-k \cdot \sigma_h)\right)^2,$$

$$q = 0.23 \cdot \sqrt{\Gamma_0} \cdot (1 - \exp(-k \cdot \sigma_h)), \Gamma_0(0) = \left|\frac{\sqrt{\dot{\varepsilon}} - 1}{\sqrt{\dot{\varepsilon}} + 1}\right|^2,$$

where $\dot{\varepsilon}$ is the surface dielectric constant; $\theta$ is the angle of incidence.

For the initial data (real part of the dielectric constant $\varepsilon = 4$, conductivity of the medium g = 0.1 S/m, wavelength $\lambda = 0.03$ m (frequency f = 10 GHz), and root mean square of the roughness height $\sigma_h^2 = 0.01$), the following dependencies, in accordance with the theoretical information given earlier, are obtained (Figure 23).

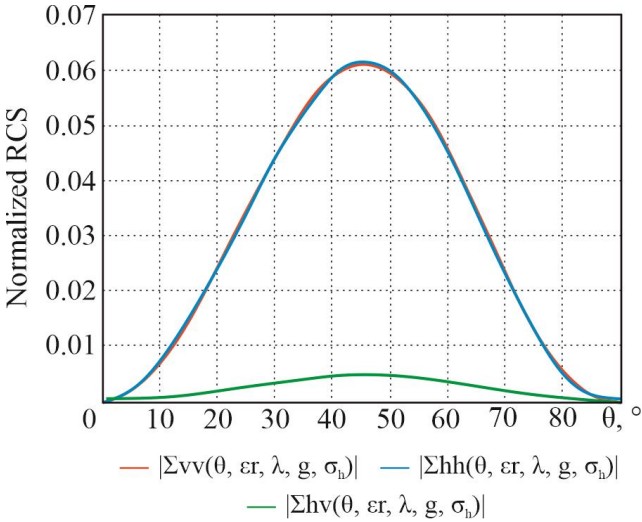

**Figure 23.** Dependences of the Oh's model normalized RCSs on the sighting angle $\theta$ at the horizontal $\Sigma$hh- and vertical $\Sigma$vv- and cross $\Sigma$hv-polarizations.

### 3.4.3. Empirical Model of a Surface with Vegetation

An empirical model of the dependence of the specific RCS of a surface with vegetation on the angle of incidence, operating frequency, radiation, and reception polarization—which works well at angles of incidence from 0° to 60° and at frequencies of 1–18 GHz—was proposed in [35] and has the following form:

$$\sigma^0(dB) = a_0 + a_1 e^{a_2\theta} + (a_3 + a_4 e^{-a_5\theta}) \cdot e^{-(a_6 - a_7\theta)f}, \tag{47}$$

where the coefficients $a_1 - a_7$ are determined by the radiation and reception polarization. The values of these coefficients are presented in Table 4.

**Table 4.** Coefficient values in Formula (47).

| Polarization | $a_0$ | $a_1$ | $a_2$ | $a_3$ | $a_4$ | $a_5$ | $a_6$ | $a_7$ |
|---|---|---|---|---|---|---|---|---|
| HH | 2.69 | −5.35 | 0.014 | −23.4 | 33.14 | 0.048 | 0.053 | 0.0051 |
| VV | 3.49 | −5.35 | 0.014 | −14.8 | 23.69 | 0.066 | 0.048 | 0.0028 |
| HV | 3.91 | −5.35 | 0.013 | −25.5 | 14.65 | 0.098 | 0.258 | 0.0021 |

The big advantage of this model is its statistical validity, since Expression (47) is derived from a large amount of data (obtained from the Skylab space station and ground measurements).

A simpler empirical model for surfaces with vegetation was also proposed in [35]:

$$\sigma^0(\text{dB}) = D + 10 \cdot \alpha \cdot \log f + 8.6 \cdot \beta \cdot f^\alpha - M \cdot \theta, \tag{48}$$

where $\alpha = 0.8$, $\beta = 0.04$, and $D = -15.5$, $M = 0.1$.

### 3.4.4. Dubois Empirical Model

The homogeneous surface model proposed by Dubois [36] agrees reasonably well with ground measurements for fields without vegetation and performs poorly on sufficiently vegetated fields. The second condition for the application of this model requires the fulfillment of the $\sigma^0_{VV} \geq \sigma^0_{HH}$ ratio. The normalized radar cross-section of such a surface at horizontal and vertical polarizations is described by the following expressions:

$$\sigma^0_{HH} = 10^{-2.75} \cdot \left( \frac{(\cos\theta)^{1.5}}{(\sin\theta)^5} \right) \cdot 10^{0.028 \cdot \varepsilon \cdot \text{tg}\theta} \cdot (k\sigma_h \sin\theta)^{1.4} \cdot \lambda^{0.7},$$

$$\sigma^0_{VV} = 10^{-2.35} \cdot \left( \frac{(\cos\theta)^3}{(\sin\theta)^3} \right) \cdot 10^{0.046 \cdot \varepsilon \cdot \text{tg}\theta} \cdot (k\sigma_h \sin\theta)^{1.1} \cdot \lambda^{0.7}, \tag{49}$$

where $\sigma_h$ is the root mean square height of the surface roughness; $\lambda$ is the wavelength. These relations are valid in the frequency range from 1.5 GHz to 11 GHz for surfaces with root mean square of the roughness heights from 0.3 cm to 3 cm and for angles of incidence $\theta = 30° \dots 65°$.

For the initial data (real part of the dielectric constant $\varepsilon = 4$, wavelength $\lambda = 0.06$ m (frequency f = 5 GHz), and root mean square of the roughness height $\sigma_h^2 = 0.02$) the following dependencies, in accordance with the theoretical information given earlier, are obtained (Figure 24).

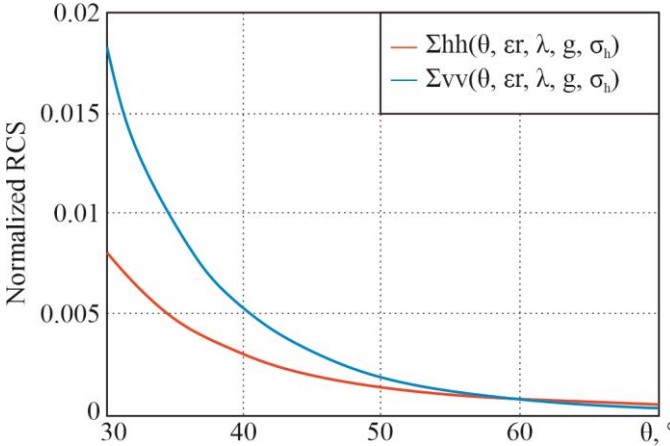

**Figure 24.** Dependences of the Dubois model's normalized RCS on the sighting angle θ at horizontal Σhh and vertical Σvv polarizations.

### 3.4.5. Model with Cylindrical Reflectors

According to this model, the surface can be described as a set of cylindrical reflectors, and thus, the normalized radar cross-section at vertical and horizontal polarizations can be represented as follows [29]:

$$\sigma_{VV}^0 = \frac{N \cdot S^2 \cdot k^2}{4 \cdot \pi} \cdot \left[\frac{(\varepsilon_1 - 1)^2 + \varepsilon_2^2}{35}\right] \cdot \frac{3 + \frac{16}{1+\varepsilon_1} + \frac{96}{(1+\varepsilon_1)^2} + \sin^2\theta \cdot \left[12 + \frac{8}{1+\varepsilon_1} - \frac{64}{(1+\varepsilon_1)^2}\right]}{\frac{3}{5}\left(\frac{d_{VV}}{k}\right)^2 + \frac{4}{5}(1 + 2\cos^2\theta)}, \tag{50}$$

$$\sigma_{HH}^0 = \frac{N \cdot S^2 \cdot k^2}{4 \cdot \pi} \cdot \frac{\left[\frac{1}{35}(\varepsilon_1 - 1)^2 + \varepsilon_2^2\right] \cdot \left[3 + \frac{16}{1+\varepsilon_1} + \frac{96}{(1+\varepsilon_1)^2}\right]}{\frac{3}{5}\left(\frac{d_{HH}}{k}\right)^2 + \frac{4}{5}(1 + 2\cos^2\theta)}, \tag{51}$$

$$d_{VV} = \frac{3}{8} \cdot N \cdot S \cdot \varepsilon_2 \sec\theta \cdot \left\{1 + \frac{12}{(1+\varepsilon_1)^2} + \sec^2\theta \cdot \left[1 - \frac{4}{(1+\varepsilon_1)^2}\right]\right\},$$

$$d_{HH} = \frac{3}{8} \cdot N \cdot S \cdot \varepsilon_2 \sec\theta \cdot \left\{1 + \frac{12}{(1+\varepsilon_1)^2}\right\},$$

$$\varepsilon = \varepsilon_1 + j\varepsilon_2,$$

where N is the number of cylinders per unit surface area, S is the cross-sectional area, and $1/d$ is the depth at which the field strength of the incident wave decreases by a factor of $e$.

For the initial data (real part of the dielectric constant $\varepsilon = 4$, conductivity of the medium g = 0.1 S/m, wavelength $\lambda = 0.03$ m (frequency f = 10 GHz), number of cylinders per unit surface area N = 100, and cross-sectional area S = 0.001) the following dependencies, in accordance with the theoretical information given earlier, are obtained (Figure 25).

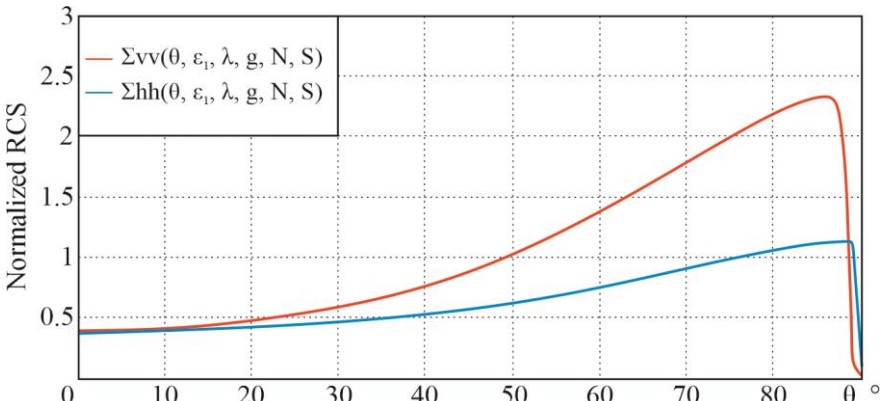

**Figure 25.** Dependence of normalized RCS for a model of cylindrical reflectors on the sighting angle $\theta$ at horizontal $\Sigma hh$ and vertical $\Sigma vv$ polarizations.

### 3.4.6. Integral Equation Model

The integral equation method (IEM), introduced in [30], is often used as an extension of the models that can determine the intensity of soil dispersion (without vegetation) at large values of the root mean square of the roughness. In the backscattering from such areas, single backscattering prevails over multiple backscattering in most cases. As a first approximation, the normalized RCS in this model is defined as follows:

$$\sigma_{PP}^0 = \frac{k^2}{2} \exp\left(-2k_z^2\sigma_h^2\right) \sum_{n=1}^{\infty} \sigma_h^{2n} \left|I_{PP}^n\right|^2 \frac{W^n(-2k_x, 0)}{n!}, \tag{52}$$

where

$$I_{PP}^n = (2k_z)^n \cdot f_{PP} \exp\left(-2k_z^2\sigma_h^2\right) + \frac{1}{2}\left\{k_z^n\left[\Gamma_{PP}(k_x, 0)\right]\right\},$$

and where the pp-index describes the type of polarization of radiation and reception.

$$f_{VV} = \frac{2\Gamma_{VV}}{\cos\theta}, \ f_{HH} = \frac{2\Gamma_{HH}}{\cos\theta},$$

where $\Gamma_{VV}$ and $\Gamma_{HH}$ are the Fresnel coefficients for vertical and horizontal polarization, respectively; $k_z = k\cos\theta$, $k_x = k\sin\theta$, and $W^n$ are the n-th-degree Fourier transforms of the surface autocorrelation function, which are equal to

$$W^n(k) = \int_0^\infty \rho^n(\xi)I_0(k\xi)d\xi,$$

where $I_0$ is a zero-order Bessel function.

For a small $k\sigma_h \ll 1$, we obtain the first-order integral equation (n = 1), for which

$$\sigma_{HH}^0 = 8k^4\sigma_h^2\left|\Gamma_{HH}(\theta)\cos^2\theta\right|^2 W(-2k_x, 0), \tag{53}$$

$$\sigma_{VV}^0 = 8k^4\sigma_h^2\left|\Gamma_{VV}(\theta)\cos^2\theta + \frac{\sin^2\theta[1+\Gamma_{VV}(\theta)]^2}{2}\left(1-\frac{1}{\dot\varepsilon}\right)\right|^2 W(-2k_x, 0). \tag{54}$$

At frequencies above 4 GHz,

$$\Gamma_{HH}(\theta) = \left|\frac{\cos\theta - \sqrt{\dot\varepsilon - \sin^2\theta}}{\cos\theta + \sqrt{\dot\varepsilon - \sin^2\theta}}\right|^2, \Gamma_{VV}(\theta) = \left|\frac{\dot\varepsilon\cos\theta - \sqrt{\dot\varepsilon - \sin^2\theta}}{\dot\varepsilon\cos\theta + \sqrt{\dot\varepsilon - \sin^2\theta}}\right|^2.$$

In the frequency range of f < 4 GHz,

$$\Gamma_{HH}(\theta) \approx \Gamma_{VV}(\theta) \approx \Gamma_0(0) = \left|\frac{\sqrt{\dot\varepsilon}-1}{\sqrt{\dot\varepsilon}+1}\right|^2,$$

where $\Gamma_0(0)$ is the Fresnel coefficient when irradiating into the nadir.

For the initial data (real part of the dielectric constant $\varepsilon = 4$, conductivity of the medium g = 0.1, wavelength $\lambda = 0.06$ m (frequency f = 5 GHz), root mean square of the roughness height $\sigma_h^2 = 10^{-5}$, and $W(-2k_x, 0) = 1$), the following dependencies, in accordance with the theoretical information given earlier, are obtained (Figure 26).

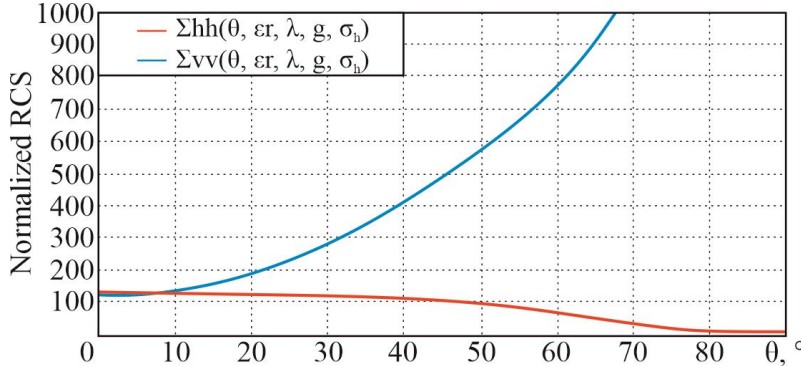

**Figure 26.** Dependence of normalized RCS for an integral equation model on the sighting angle $\theta$ at horizontal $\Sigma hh$ and vertical $\Sigma vv$ polarizations at frequencies above 4 GHz.

For the initial data (real part of the dielectric constant $\varepsilon = 4$, conductivity of the medium g = 0.1, wavelength $\lambda = 0.3$ m (frequency f = 1 GHz), root mean square of the roughness height $\sigma_h^2 = 10^{-4}$, and $W(-2k_x, 0) = 1$), the following dependencies are obtained (Figure 27).

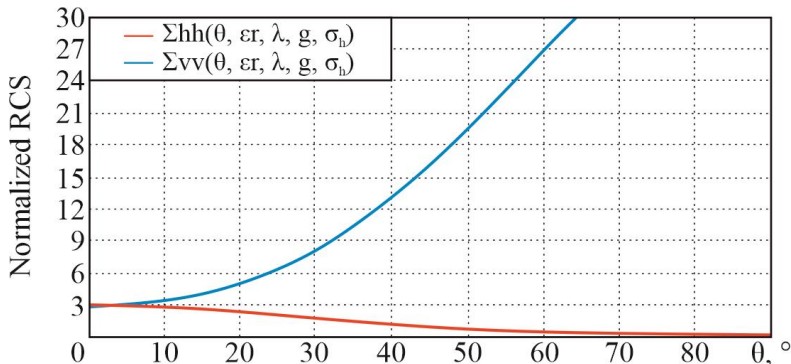

**Figure 27.** Dependence of normalized RCS for an integral equation model on the sighting angle θ at horizontal Σhh and vertical Σvv polarizations at frequencies less than 4 GHz.

### 3.4.7. Model with Near-Surface Wind

According to this model, the backscattering cross-section is connected to the near-surface wind speed as follows [37]:

$$\sigma_k^0 = \alpha_{0k}\nu^{\gamma_{0k}} + \alpha_{1k}\nu^{\gamma_{1k}}\cos\varphi + \alpha_{2k}\nu^{\gamma_{2k}}\cos 2\varphi, \tag{55}$$

where $k = (VV, HH)$, $\nu$ is the wind speed [m/s], and $\varphi$ is the angle in the opposite direction of the wind vector [deg].

The best estimates of the constants in this expression are shown in Table 5.

**Table 5.** Constant values for the model.

| k | φ, deg | $\alpha_0$ | $\gamma_0$ | $\alpha_1$ | $\gamma_1$ | $\alpha_2$ | $\gamma_2$ |
|---|---|---|---|---|---|---|---|
| | 30 | $8.4 \times 10^{-4}$ | 1.85 | $5.3 \times 10^{-5}$ | 1.76 | $3.3 \times 10^{-4}$ | 1.95 |
| VV | 40 | $1.3 \times 10^{-4}$ | 2.15 | $3.5 \times 10^{-5}$ | 2.03 | $6.4 \times 10^{-5}$ | 2.27 |
| | 50 | $4.2 \times 10^{-4}$ | 2.34 | $1.6 \times 10^{-5}$ | 2.22 | $2.0 \times 10^{-5}$ | 2.46 |
| | 30 | $1.2 \times 10^{-4}$ | 1.62 | $2.64 \times 10^{-4}$ | 1.54 | $3.8 \times 10^{-4}$ | 1.7 |
| HH | 40 | $7.6 \times 10^{-4}$ | 2.05 | $3.9 \times 10^{-5}$ | 1.94 | $2.8 \times 10^{-5}$ | 2.16 |
| | 50 | $9.6 \times 10^{-4}$ | 2.40 | $7.2 \times 10^{-6}$ | 2.28 | $3.9 \times 10^{-6}$ | 2.54 |

Based on the above-mentioned information, the following dependencies are obtained (Figure 28).

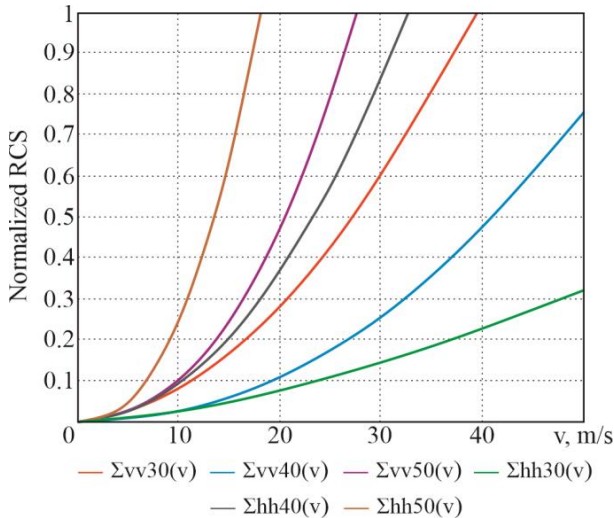

**Figure 28.** Dependences of normalized RCSs of wind speed $\nu$ for angles $\varphi = 30, 40, 50$ deg. at horizontal Σhh30, Σhh40, Σhh50 and vertical Σvv30, Σvv40, Σvv50 polarizations.

3.4.8. Shi's Model Algorithm

The relations of normalized RCSs according to this algorithm are described by three equations [38]:

$$\sigma_{pp}^0 = |\alpha_{pp}|^2 \left[ \frac{S_R}{a_{pp}(\theta) + b_{pp}(\theta)S_R} \right],$$

$$10\lg\left[\frac{|\alpha_{VV}|^2 + |\alpha_{HH}|^2}{\sigma_{VV}^0 + \sigma_{HH}^0}\right] = a_{VH}(\theta) + b_{VH}(\theta)10\lg\left[\frac{|\alpha_{VV}||\alpha_{HH}|}{\sqrt{\sigma_{VV}^0 \sigma_{HH}^0}}\right], \quad (56)$$

$$\frac{\sigma_{HH}^0}{\sigma_{VV}^0} = \frac{|\alpha_{HH}|^2}{|\alpha_{VV}|^2} \cdot \exp[a_r(\theta) + k \cdot (b_r(\theta) + c_r(\theta)) \cdot W],$$

where pp are the indices describing the type of polarization, $S_R = (k\sigma_h)^2 W$, and W are the Fourier transforms of the spatial correlation function of the surface. All the coefficients in these equations are functions of the angle of incidence $\theta$ only and are given in [38] in polynomial form.

3.4.9. Empirical Model of Backscattering from Snow

In the millimeter range, backscattering from snow includes surface backscattering from air–snow and snow–ground interfaces, as well as volumetric backscattering from ice crystals in the snow layer [31,39]. The specific EPR of snow is determined by the following factors: radar frequency, transmission and reception polarization, radar beam grazing angle, and electrophysical and geometric characteristics of the snow cover.

The main snow parameter that determines the normalized RCS is its water equivalent,

$$W = \rho_S h,$$

where $\rho_S$ is the snow density, and h is the height of the snow cover.

In general, the normalized RCS for snow can be described as in [31,39]:

$$\sigma^0 = \sigma_{ss}^0(\lambda, \theta) + \sigma_s^0(\theta') + \frac{\gamma_{sa}^2(\theta')}{L^2(\theta')} \cdot \sigma_{soil}^0(\theta'), \quad (57)$$

where $\sigma_{ss}^0$ is the normalized RCS of the air–snow interface, $\sigma_s^0(\theta')$ is the normalized RCS from the volumetric snow layer, $\gamma_{sa}^2(\theta')$ is the reactive power factor of the air–snow interface, and $\sigma_{soil}^0(\theta')$ is the normalized RCS of the soil.

The angle $\theta'$ can be defined through the angle $\theta$ in the following expression:

$$\sin\theta = \sqrt{\varepsilon_s} \cdot \sin\theta',$$

where $\varepsilon_s$ is the dielectric constant of snow.

*3.5. Model Selection Algorithm*

Based on the analysis and processing of a large amount of information following the proposed classification, an algorithm for surface model selection [6,46] was created (Figure 29), and a software product prototype for selecting a surface model type was created (Figure 30). Thus, the user will be able to select functional dependencies for certain experiments or calculations based on known initial data (system type, operating frequencies, and physical and geometric characteristics of the surface).

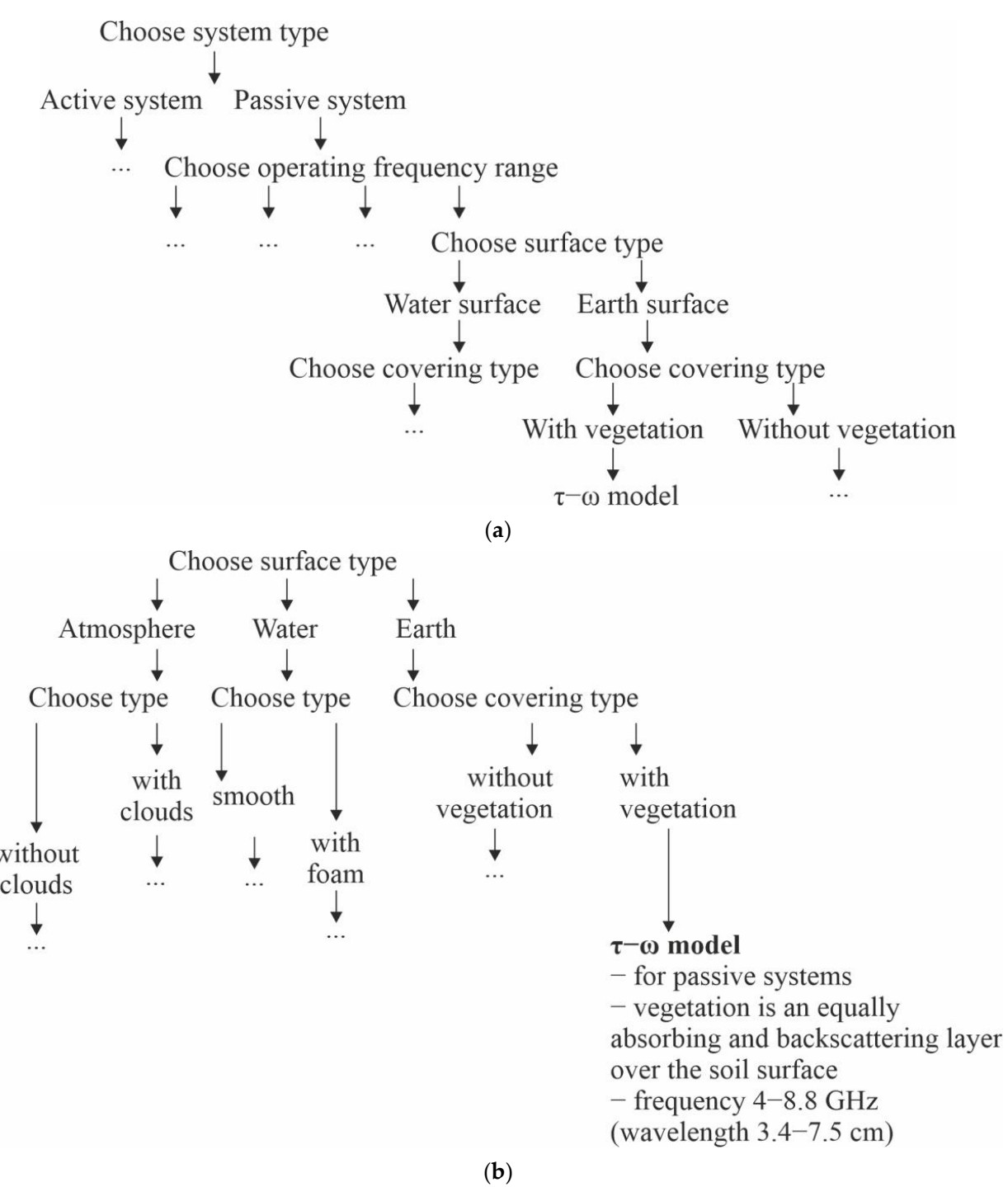

(**a**)

(**b**)

**Figure 29.** Model selection algorithms: (**a**) For a known system type; (**b**) For a known studied surface type.

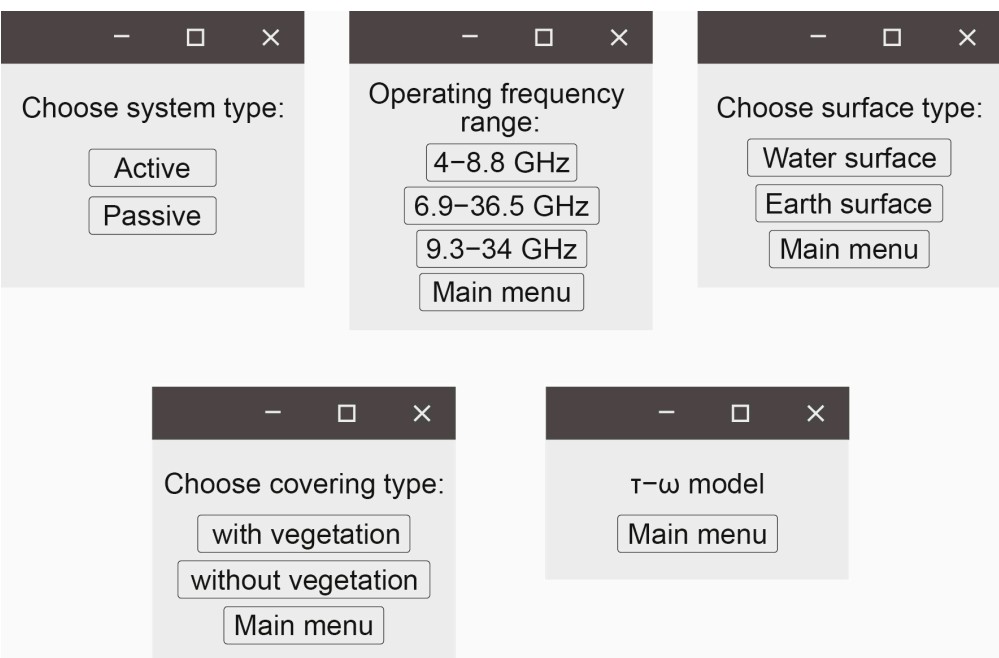

**Figure 30.** Fragment of the software interface.

At the first stage of the research, it is necessary to determine the initial conditions: whether the researcher has information about the system parameters or the type of surface. If the problem statement contains the system parameters and technical characteristics and it is necessary to establish its capabilities (variants of the surfaces), the first algorithm is recommended for use (Figure 29a). If the research goal is to study a specific type of surface (its geometric and physical characteristics are at least partially known), a second approach is used (Figure 29b), which allows us to select options for the research's technical implementation.

Next, you need to select the appropriate parameters of the system or surface and, as a result, you will receive recommendations on how to use the surface model. Despite its simplicity, this algorithm will simplify and speed up some stages of Earth's surface remote sensing.

Due to the fact that the field of remote sensing is rapidly evolving—new models are appearing, existing ones are being improved, and various experimental studies are being conducted—the information table and the proposed algorithm can be supplemented and improved.

At this stage, the algorithm is implemented as software (written in Python), and no study of the software's reliability has been conducted. If the proposed algorithm is of interest to specialists in surface remote sensing, further development of this work towards the creation of software with the corresponding necessary research and analysis is possible.

## 4. Discussion

Nowadays, the task of measuring the parameters of various types of surfaces, the Earth's surfaces in particular, is very relevant, both from a scientific point of view and for practical applications in agriculture, ecology, geology, and other fields. When preparing and designing practical experiments and analyzing the results of modeling or specific experiments, the interpretation task must be solved. Moreover, the accuracy and validity of the results are determined by the relevance of the model that describes the studied surface and relates the surface parameters to the signals that are recorded by the radio system. Thus, the authors performed a thorough analysis of open sources regarding different types of such models. This paper provides a comprehensive description and analysis of such

models that can be used under different conditions to describe various types of surfaces: earth surface, vegetated surface, sea surface, and others.

In addition, in this paper and on the basis of this material, an algorithm for selecting the optimal option is proposed, which can and should undoubtedly be expanded when dealing with new models of the relation of surface parameters to signals received by radio engineering systems. As a further development, we can also consider refining the software application prototype, improving the ergonomics and clarity of information perception.

It is clear that the diversity of our planet's coverings is not limited to the types of surfaces presented in this paper, e.g., there are phenomenological models, facet models, and more. Also, there may be cases when the illuminated area of the observed surface contains a diverse terrain. It should be noted that the result of the experiment largely depends on the radio system's resolution, but the task of achieving high resolution is a technical task that requires separate solutions and approaches.

Thus, as numerous studies show, taking into account even the simplest dependencies of received fields or their statistical characteristics with the surface structure and its electrophysical parameters brings a lot of additional information. It gives an opportunity, by recording radio and radio-thermal fields of different polarizations, to obtain information about dielectric permittivity, conductivity, and surface geometric characteristics, allowing us to determine the temperature, humidity, salinity, surface density, roughness height profiles, and more.

Results achieved in the field of active and passive radar designs with optimal spatial and temporal processing of registered electromagnetic fields and high resolutions allow us to set inverse problems of high-precision estimations of surface parameters and their mapping using data of electrodynamic models and, to a great extent, stimulate the development of new methods of solving direct electrodynamic problems in modeling more complex surfaces and Earth's real covers.

## 5. Conclusions

This paper presents a comparative analysis of the existing models for self-radiation signals or signals reflected by a surface (with surface characteristics). On this basis, a relation models' classifier is proposed. For some of the most commonly used types of surfaces, theoretical information and simulation results are presented, which demonstrate the properties of the models. Following the proposed classification, an algorithm for selecting a model based on known initial data is constructed, and a software product that implements this algorithm is created. The obtained results are recommended for use in planning practical experiments on Earth surface remote sensing, in interpreting the results of such experiments, in modeling similar studies, as well as for educational purposes to ensure a better understanding of theoretical material.

**Author Contributions:** Conceptualization, V.V. and K.N.; methodology, V.V.; software, K.N.; validation, K.N., K.B., D.K. and G.C.; formal analysis, V.V.; investigation, K.N.; resources, D.K. and K.B.; data curation, D.K. and G.C.; writing—original draft preparation, K.N.; writing—review and editing, D.K.; visualization, G.C.; supervision, V.V. and K.B.; project administration, K.N. and D.K. All authors have read and agreed to the published version of the manuscript.

**Funding:** This research received no external funding.

**Data Availability Statement:** The original contributions presented in the study are included in the article, further inquiries can be directed to the corresponding author.

**Conflicts of Interest:** The authors declare no conflicts of interest.

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
