# Peer review of "Relation Models of Surface Parameters and Backscattering (or Radiation) Fields as a Tool for Solving Remote Sensing Problems"

_computation, doi:10.3390/computation12050104_

Round 1
Reviewer 1 Report
Comments and Suggestions for Authors
Please find the attached report.

Reviewer 2 Report
Comments and Suggestions for Authors
I have read with interest the manuscript entitled:
Relation Models of Surface Parameters and Backscattering (or Radiation) Fields as a Tool for Solving Remote Sensing Problems.
I appreciate its high substantive level. I have no comments about the scientific aspect. I was surprised that a large group of authors provided rich bibliographies (many citations). Among these works, there is only one work whose author is only one of the authors from this large group. In addition, this is a work from 18 years ago. Do the authors have no published work and no achievements they would like to boast about? Where did this job suddenly come from? There is only one information on the Internet about a publication whose subject matter is identical to that in the submitted manuscript. However, this material is not publicly available and that's why I have questions:
1. why the paper “Algorithm for Selecting a Surface Model for Remote Sensing of Earth's Surfaces” DOI: 10.1109/ELIT61488.2023.10310806 is not cited in the manuscript;
2. to what extent does the manuscript contain new content and to what extent have the analyzes provided previously been published? Can you please send it to me?
Other comments: good language level, but there are corrections to be made
They can be found in the attachment, which contains a scan with manual corrections.
With kind regards,
X

Some corrections must be done. English language is quite good.
Round 2
Reviewer 1 Report
Comments and Suggestions for Authors
All the suggested changes have been incorporated in the revised version.
Reviewer 2 Report
Comments and Suggestions for Authors
Dear Authors,
"We are sorry to say that due to the journals copyright restrictions we cannot share our previous papers."
I do not think it is true. Hard to believe that such a restriction is related to a conference paper. Anyway, I have got that paper. I have not noticed a self plagiarism.
I won't block this manuscript, but I don't enjoy with its form: a lot of textbook knowledge, scant information about the relevant research (including lack of clarity about what was and what has been done), only a few paragraphs of fuzzy conclusions.